# On Second Order Behaviour
# in Augmented Neural ODEs

**Alexander Norcliffe**
Department of Physics
University of Cambridge
alex.norcliffe98@gmail.com

**Cristian Bodnar**\*, **Ben Day**\*, **Nikola Simidjievski, Pietro Liò**
Department of Computer Science and Technology
University of Cambridge
{cb2015, bjd39, ns779, pl219}@cam.ac.uk

## Abstract

Neural Ordinary Differential Equations (NODEs) are a new class of models that transform data continuously through infinite-depth architectures. The continuous nature of NODEs has made them particularly suitable for learning the dynamics of complex physical systems. While previous work has mostly been focused on first order ODEs, the dynamics of many systems, especially in classical physics, are governed by second order laws. In this work, we consider Second Order Neural ODEs (SONODEs). We show how the adjoint sensitivity method can be extended to SONODEs and prove that the optimisation of a first order coupled ODE is equivalent and computationally more efficient. Furthermore, we extend the theoretical understanding of the broader class of Augmented NODEs (ANODEs) by showing they can also learn higher order dynamics with a minimal number of augmented dimensions, but at the cost of interpretability. This indicates that the advantages of ANODEs go beyond the extra space offered by the augmented dimensions, as originally thought. Finally, we compare SONODEs and ANODEs on synthetic and real dynamical systems and demonstrate that the inductive biases of the former generally result in faster training and better performance.

## 1 Introduction

Residual Networks (ResNets) [8] have been an essential tool for scaling the capabilities of neural networks to extreme depths. It has been observed that the skip layers that these networks employ can be seen as an Euler discretisation of a continuous transformation [7, 12, 19]. Neural Ordinary Differential Equations (NODEs) [3] are a new class of models that consider the limit of this discretisation step, naturally giving rise to an ODE that can be optimised via black-box ODE solvers. Their continuous depth makes them particularly suitable for learning and modelling the unknown dynamics of complex systems, which often cannot be described analytically.

Since the introduction of NODEs, many variants have been proposed [4, 10, 14, 17, 20, 22, 24]. While a few of these models use second order dynamics [14, 17, 24], no in-depth study on second order behaviour in Neural ODEs exists even though most dynamical systems that arise in science, such as Newton's equations of motion and oscillators, are governed by second order laws. To fill this void, we consider Second Order Neural ODEs (SONODEs) and second order dynamics for the broader class of

---

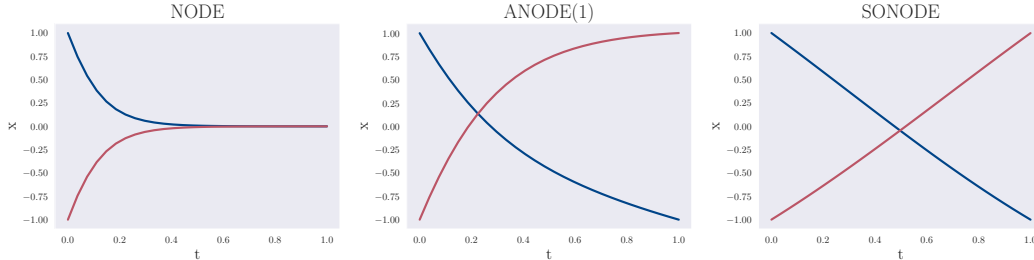

Figure 1: Three learnt trajectories from the one-dimensional compact parity experiment (originally named $g_{1d}$ in Dupont et al. [4]). NODEs, as expected, are not able to learn the mapping *(left)*, ANODE(1) is able to learn it *(middle)*, and SONODEs learn the simplest trajectory given by the solution in Equation (7) *(right)*.

models formed by Augmented Neural ODEs (ANODEs). Unlike previous approaches, which mainly focus on classification tasks, we use low-dimensional physical systems, often with known analytic solutions, as our main arena of investigation. As we will show, the simplicity of these systems is useful in analysing the properties of these models.

To summarise our contributions, we begin by studying more closely the optimisation of SONODEs by generalising the adjoint sensitivity method to second order models. We continue by analysing how some of the properties of ANODEs extend to SONODEs and show that the latter can often find simpler solutions for the problems we consider. Our analysis also extends to ANODEs and demonstrates that they are capable of learning higher-order dynamics, sometimes with just a few additional dimensions. However, the way they do so has deeper implications for their functional loss landscape and their interpretability as a scientific tool. Finally, we compare SONODEs and ANODEs on real and synthetic second order dynamical systems. Our results reveal that the inductive biases in SONODEs are beneficial in this setting. Our code is available online at https://github.com/a-norcliffe/sonode.

## 2   Background

As discussed in the introduction, Neural ODEs (NODEs) can be seen as a continuous variant of ResNet models [8], whose hidden state evolves continuously according to a differential equation

$$\dot{\mathbf{x}} = f^{(v)}(\mathbf{x}, t, \theta_f), \qquad \mathbf{x}(t_0) = \mathbf{X}_0, \tag{1}$$

whose velocity is described by a neural network $f^{(v)}$ with parameters $\theta_f$ and initial position given by the points of a dataset $\mathbf{X}_0$. As shown by Chen et al. [3], the gradients can be computed through an abstract adjoint state $\mathbf{r}(t)$, once its dynamics are known.

Our investigations are mainly focused on Augmented Neural ODEs (ANODEs) [4], which append states $\mathbf{a}(t)$ to the ODE:

$$\mathbf{z} = \begin{bmatrix} \mathbf{x} \\ \mathbf{a} \end{bmatrix}, \quad \dot{\mathbf{z}} = f^{(v)}(\mathbf{z}, t, \theta_f), \quad \mathbf{z}(t_0) = \begin{bmatrix} \mathbf{X}_0 \\ g(\mathbf{X}_0, \theta_g) \end{bmatrix}. \tag{2}$$

We note that, unlike the original formulation, we allow for the initial values of the augmented dimensions $\mathbf{a}(t_0)$ to be learned as a function of $\mathbf{x}(t_0)$ by a neural network $g$ with parameters $\theta_g$. For the remainder of the paper, we use the ANODE($D$) notation to signify the use of $D$ augmented dimensions.

We are almost exclusively concerned with the problem of learning and modelling the behaviour of dynamical systems, given $N + 1$ sample points $\mathbf{X}_{t \in T}$, $t = (t_0, \dots, t_N)$, from a fixed set of its trajectories at multiple time steps included in the set $T$. For such tasks, we use the mean squared error (MSE) between these points and the corresponding predicted location over all time steps for training the models. For the few toy classification tasks we include, we optimise only for the linear separability of the final positions via the cross-entropy loss function.

# 3 Second Order Neural Ordinary Differential Equations

We consider Second Order Neural ODEs (SONODEs), whose initial position $\mathbf{x}(t_0)$, initial velocity $\dot{\mathbf{x}}(t_0)$, and acceleration $\ddot{\mathbf{x}}$ are given by

$$\mathbf{x}(t_0) = \mathbf{X}_0, \qquad \dot{\mathbf{x}}(t_0) = g(\mathbf{x}(t_0), \theta_g), \qquad \ddot{\mathbf{x}} = f^{(a)}(\mathbf{x}, \dot{\mathbf{x}}, t, \theta_f), \qquad (3)$$

where $f^{(a)}$ is a neural network with parameters $\theta_f$. Alternatively, SONODEs can be seen as a system of coupled first-order neural ODEs with state $\mathbf{z}(t) = [\mathbf{x}(t), \mathbf{a}(t)]$:

$$\mathbf{z} = \begin{bmatrix} \mathbf{x} \\ \mathbf{a} \end{bmatrix}, \quad \dot{\mathbf{z}} = f^{(v)}(\mathbf{z}, t, \theta_f) = \begin{bmatrix} \mathbf{a} \\ f^{(a)}(\mathbf{x}, \mathbf{a}, t, \theta_f) \end{bmatrix}, \quad \mathbf{z}(t_0) = \begin{bmatrix} \mathbf{X}_0 \\ g(\mathbf{X}_0, \theta_g) \end{bmatrix}. \qquad (4)$$

This formulation makes clear that SONODEs are a type of ANODE with constraints on the structure of $f^{(v)}$, and offers a way to reuse NODE's first order adjoint method [3] for training, as in previous work [14, 24]. However, a pair of questions remain about the optimisation of SONODEs: firstly, what is the ODE that the second order adjoint follows? And, consequently, how does the second order adjoint sensitivity method compare with first order adjoint-based optimisation? To address these questions, we show how the adjoint sensitivity method can be generalised to SONODEs.

**Proposition 3.1.** *The adjoint state* $\mathbf{r}(t)$ *of SONODEs follows the second order ODE*

$$\ddot{\mathbf{r}} = \mathbf{r}^T \frac{\partial f^{(a)}}{\partial \mathbf{x}} - \dot{\mathbf{r}}^T \frac{\partial f^{(a)}}{\partial \dot{\mathbf{x}}} - \mathbf{r}^T \frac{d}{dt}\left( \frac{\partial f^{(a)}}{\partial \dot{\mathbf{x}}} \right) \qquad (5)$$

The proof and boundary conditions for this ODE is given in Appendix B. As an additional contribution, we include an alternative proof to those of Chen et al. [3] and Pontryagin [18] for the first order adjoint. Given that the dynamics of the abstract adjoint vector are known, its state at all times $t$ can be used to train the parameters $\theta_f$ using the integral

$$\frac{dL}{d\theta_f} = -\int_{t_n}^{t_0} \mathbf{r}^T \frac{\partial f^{(a)}}{\partial \theta_f} dt, \qquad (6)$$

where $L$ denotes the loss function and $t_n$ is the timestamp of interest. The gradient with respect to the parameters of the initial velocity network, $\theta_g$, can be found in Appendix B. To answer the second question, we compare this gradient against that obtained through the adjoint of the first order coupled ODE from Equation (4).

**Proposition 3.2.** *The gradient of* $\theta_f$ *computed through the adjoint of the coupled ODE from (4) and the gradient from (6) are equivalent. However, the latter requires at least as many matrix multiplications as the former.*

This result motivates the use of the first order coupled ODE as it presents computational advantages. The proof in Appendix B shows that this is due to the dynamics of the adjoint from the coupled ODE, which contain entangled representations of the adjoint. This is in contrast to the disentangled representation in Equation (5), where the adjoint state and velocity are separated. It is the entangled representation that permits the faster computation of the gradients for the coupled ODE. We will see in Section 5.3 that entangled representations in ANODEs are a reoccurring phenomenon, and their effects are not always beneficial, as in this case. We use the first order ODE optimisation for the remainder of our experiments.

# 4 Properties of SONODEs

In this section, we analyse certain properties of SONODEs and illustrate them with toy examples.

## 4.1 Generalised parity problem

It is known that unique trajectories in NODEs cannot cross at the same time [4, 14]. We extend this to higher order Initial Value Problems (IVP). Proofs are presented in Appendix A.

**Proposition 4.1.** *For a k-th order IVP, if the k-th derivative of* $\mathbf{x}$ *is Lipschitz continuous and has no explicit time dependence, then unique phase space trajectories cannot intersect at an angle. Similarly, a single phase space trajectory cannot intersect itself at an angle.*

While this shows SONODE trajectories cannot cross in phase space, they can cross in real space if they have different velocities. To illustrate this, we introduce a generalised parity problem, an extension to $D$ dimensions of the $g_{1d}$ function from Dupont et al. [4], which maps $\mathbf{x} \rightarrow -\mathbf{x}$. We remark that SONODEs should be able to learn a parity flip in any number of dimensions, with a trivial solution

$$f^{(a)}(\mathbf{x}, \dot{\mathbf{x}}, t, \theta_f) = 0, \qquad g(\mathbf{x}(t_0), \theta_g) = -\frac{2}{t_N - t_0}\mathbf{x}(t_0) \qquad (7)$$

This is equivalent to all points moving in straight lines through the origin to $-\mathbf{x}(t_0)$. We first visualise the learnt transformation in the one dimensional case (Figure 1), for points initialised at $\pm 1$. SONODEs learn the simplest trajectories for this problem.

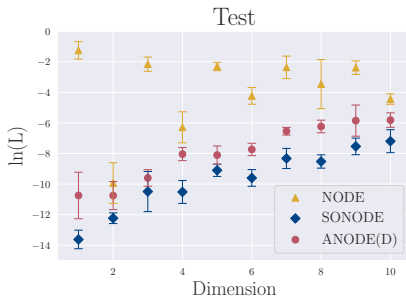

For higher dimensions, we first remark that NODEs are able to produce parity flips for even dimensions by pairing off the dimensions and performing a $180°$ rotation in each pair. This solution does not apply to odd-dimensional cases because there is always an unpaired dimension that is not rotated. In addition to the dimensional-parity effect, as volume increases exponentially with the dimensionality, the density exponentially decreases (given the number of points in the dataset remains constant). This makes it easier to manipulate the points without trajectories crossing, and so, it is expected that the problem will become easier for NODEs as dimensionality increases.

Figure 2: The logarithm of the loss in each dimension for the generalised parity problem. SONODE has the lowest loss, while the NODE loss generally oscillates between dimensions as predicted.

In Figure 2, we investigate parity flips in higher dimensions, using 50 training points and 10 test points, randomly generated between -1 and 1 in each dimension. For NODEs, as predicted, the loss oscillates over dimensions and, for odd dimensions, the loss decreases with the number of dimensions. ANODEs perform better than NODES, especially in odd dimensions, where it can rotate the unpaired dimension through the additional space. SONODEs have the lowest loss in every generalisation, which can be associated with the existence of the trivial solution in any number of dimensions, given by Equation (7).

### 4.1.1 Nested n-spheres

Dupont et al. [4] prove that a transformation under NODEs has to be a homeomorphism, preserving the topology of the input space, and as such, they cannot learn certain transformations. Similarly to ANODEs, SONODEs avoid this problem.

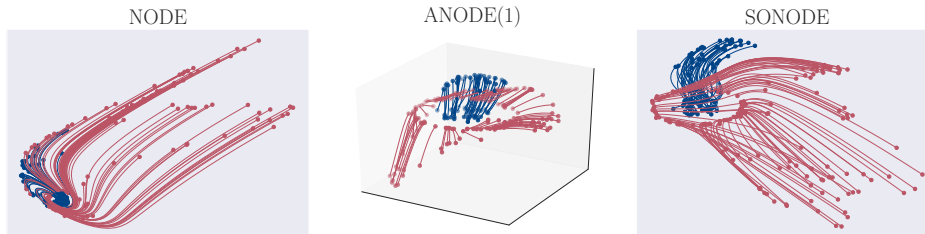

Figure 3: The trajectories learnt by NODE *(left)*, ANODE *(middle)* and SONODE *(right)* for the nested n-spheres problem in 2D. NODE preserves the topology, so the blue region cannot escape the red region. ANODE, as expected, uses the third dimension to separate the two regions. For SONODE, the points pass through each other in real space.

**Proposition 4.2.** *SONODEs are not restricted to homeomorphic transformations in real space.*

The proof can be found in Appendix C. To illustrate this, we perform an experiment on the nested n-spheres problem [4], (the name is taken from [14], originally called $g$ function [4]), where the elements of the blue class are surrounded by the elements of the red class (Figure 3) such that a homeomorphic transformation in that space cannot linearly separate the two classes. As expected, only ANODEs and SONODEs can learn a mapping.

## 5 Second order behaviour in SONODEs and ANODEs

Previously, the benefits of ANODEs were attributed only to the extra space they have in which to move [4]. However, in this section, we show that coupled first order ODEs, such as ANODEs, are also able to represent higher-order order behaviour. Additionally, we study the functional forms ANODEs can use to learn this. Unless stated, we consider ANODEs in their original formulation where $\mathbf{a}(t_0) = 0$.

### 5.1 How do ANODEs learn second order dynamics?

Consider a SONODE as in Equation (3). Similarly to the coupled ODE from Equation (4), ANODEs can represent this if the state, $\mathbf{z} = [\mathbf{x}, \mathbf{a}]$, is augmented such that $\mathbf{a}$ has the same dimensionality as $\mathbf{x}$:

$$\mathbf{z}(t_0) = \begin{bmatrix} \mathbf{x}(t_0) \\ 0 \end{bmatrix}, \quad \dot{\mathbf{z}} = \begin{bmatrix} \mathbf{a} + \dot{\mathbf{x}}(t_0) \\ f^{(a)}(\mathbf{x}, \dot{\mathbf{x}}, t, \theta_f) \end{bmatrix} = \begin{bmatrix} \mathbf{a} + g(\mathbf{x}(t_0), \theta_g) \\ f^{(a)}(\mathbf{x}, \mathbf{a} + g(\mathbf{x}(t_0), \theta_g), t, \theta_f) \end{bmatrix}, \quad (8)$$

where $\mathbf{a}$ differentiates to the acceleration and, because $\mathbf{a}(t_0) = 0$, the initial velocity is added to it to obtain the correct dynamics. Generalising this, it is clear to see how ANODEs can also learn $k$-th order ODEs, by splitting up the augmented part $\mathbf{a}$ into $k-1$ vectors with the same dimensionality as $\mathbf{x}$. However, if ANODEs were to learn higher order dynamics this way, $\mathbf{x}(t_0)$ is required as an input, just as in data-controlled neural ODEs [14]. To show this is not usually the case, we let ANODE(1) learn two 1D functions at the same time with a shared ODE, using the same set of parameters, but different initial conditions. Specifically, we consider two damped harmonic oscillators

$$x_1(t) = e^{-\gamma t}\sin(\omega t), \qquad x_2(t) = e^{-\gamma t}\cos(\omega t) \qquad (9)$$

where $\gamma$ can be zero so that there is no decay.

SONODEs can learn these using the functional form

$$f^{(a)}(x, \dot{x}, t, \theta_f) = -(\omega^2 + \gamma^2)x - 2\gamma\dot{x}, \qquad g(x(0), \theta_g) = -(\omega + \gamma)x(0) + \omega \qquad (10)$$

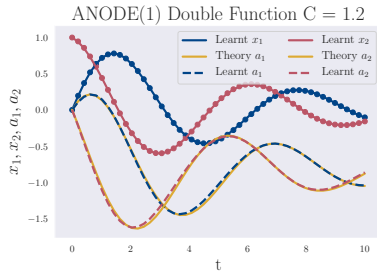

Figure 4: ANODE(1) learning two functions using the same parameters, for $\omega = 1$ and $\gamma = 0.1667$. The real trajectories are going through their sampled data points. Augmented trajectories are plotted over their theoretical trajectories given by Equation (11) for $C = 1.2$.

It is not immediately obvious how ANODEs could solve this, especially if they follow Equation (8), where $x(t_0)$ is needed as an input to determine $\dot{x}(t_0)$. However, Figure 4 shows that ANODEs are able to fit the two functions in the same training session. We observe that ANODEs approximate a solution of the form:

$$\begin{bmatrix} \dot{x} \\ \dot{a} \end{bmatrix} = \begin{bmatrix} Ca - \omega x - \gamma x + \omega \\ \omega a - \gamma a - \frac{1}{C}(2\omega^2 x + \gamma\omega - \omega^2) \end{bmatrix} \qquad (11)$$

Using $a(0) = 0$, this gives the correct ODE and initial conditions in Equation (10), for all finite, non-zero $C$.

We remark that the state $x$ and the augmented dimension $a$ are entangled in the velocity of the state and $\dot{x} \neq a$. This example gives an intuition about the way ANODEs can learn second order behaviour through an ODE as in Equation (11). We now formalise this intuition and give a general expression:

**Proposition 5.1.** *The general form ANODEs learn second order behaviour is given by:*

$$\begin{bmatrix} \dot{\mathbf{x}} \\ \dot{\mathbf{a}} \end{bmatrix} = \begin{bmatrix} F(\mathbf{x}, \mathbf{a}, t, \theta_F) \\ G(\mathbf{x}, \mathbf{a}, t, \theta_G) \end{bmatrix}, \qquad G = \left(\frac{\partial F}{\partial \mathbf{a}^T}\right)^{-1}_{left}\left(f^{(a)} - \frac{\partial F}{\partial \mathbf{x}^T}F - \frac{\partial F}{\partial t}\right) \qquad (12)$$

This result is derived in Appendix D. It shows that SONODEs and ANODEs learn second order dynamics in different ways. ANODEs learn an abstract function $F$ that at $t_0$ is equal to the initial velocity, and another function $G$ that couples to $F$ giving it the right acceleration. In contrast, SONODEs are constrained to learn the acceleration and initial velocity directly. This also leads to several useful properties that we investigate next.

## 5.2 Minimal augmentation

The first property we analyse is called *minimal augmentation*. It refers to the fact that ANODEs can learn second order dynamics even when the number of extra dimensions is less than the dimensionality of the real space.

**Corollary 5.1.1.** *When the system from Proposition 5.1 is overdetermined (i.e. $dim(\mathbf{a}) < dim(\mathbf{x})$) and has a solution, the Moore-Penrose left pseudo-inverse produces that solution, given by $G$. If no solution exists, $G$ is the best least-squares approximation.*

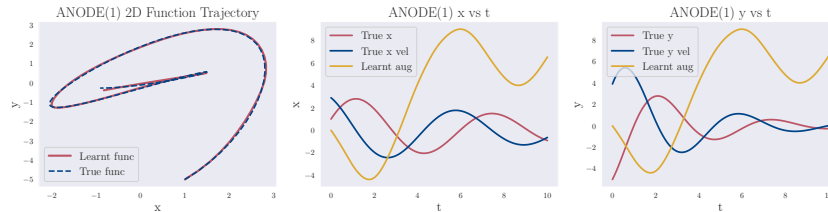

Figure 5: ANODE(1) fitting a 2D second order function: real plane view *(left)*, individual dimensions *(middle, right)*. ANODEs do not always need double the dimensions to learn second order.

In effect, ANODE is learning a system of linear equations parametrised by deep neural networks. To learn second order dynamics with minimal augmentation, it must learn an overdetermined linear system allowing a solution. Depending on the form of $\ddot{\mathbf{x}}$, it is possible that an $F$ with explicit $\mathbf{a}$ dependence that produces a degenerate system like this could be learned. In turn, this would allow a complementary $G$ to be learned. In fact, systems like this can naturally arise when the dynamics are latent and lower-dimensional and many of the observed dimensions become redundant. For instance, two spatial dimensions suffice for a pendulum moving in a plane of the 3D space.

However, even if an overdetermined system allowing a solution could not be learned due to the additional constraints acting on $F$, the left Moore-Penrose pseudo-inverse from Proposition 5.1 would still produce a $G$ that is a best least-squares approximation. If the matrix $A = \frac{\partial F}{\partial \mathbf{a}^T}$ has full rank, then the left inverse is given by $(A^T A)^{-1} A^T$. In general, the closer $dim(\mathbf{a})$ gets to $dim(\mathbf{x})$, the better this approximation will be.

To demonstrate minimal augmentation, we consider a two dimensional second order ODE, whose starting conditions and respective $\omega$'s and $\gamma$'s were chosen randomly such that

$$\begin{bmatrix} \ddot{x} \\ \ddot{y} \end{bmatrix} = \begin{bmatrix} -(\omega_x^2 + \gamma_x^2)x - 2\gamma_x \dot{x} \\ -(\omega_y^2 + \gamma_y^2)y - 2\gamma_y \dot{y} \end{bmatrix}, \qquad \begin{bmatrix} x \\ y \end{bmatrix} = \begin{bmatrix} e^{-0.1t}(3\sin(t) + \cos(t)) \\ e^{-0.3t}(2\sin(1.2t) - 5\cos(1.2t)) \end{bmatrix} \qquad (13)$$

ANODE(1) is able to learn this function, as shown in Figure 5. Moreover, the augmented dimension trajectory differs greatly from the velocity of the ODE in either of the two spatial dimensions.

## 5.3 Interpretability of ANODEs

The result from Proposition 5.1 also raises the issue of how *interpretable* ANODEs are. For example, when investigating the dynamics of physical systems it is useful to know the force equation. This is straightforward with SONODEs, which directly learn the acceleration as a function of position, velocity and time. However, ANODEs learn the dynamics through an abstract alternative ODE where the state and augmented dimensions are entangled. This is similar to the widely studied problem of entangled representations [1, 9, 15].

We then train both ANODE(2) and SONODE to learn the dynamics of the ODE from Equation (13), and provide them both with the correct initial velocity. Figure 6 shows the results for two different runs for both models. Though ANODE(2) is able to learn the true trajectory in real space,

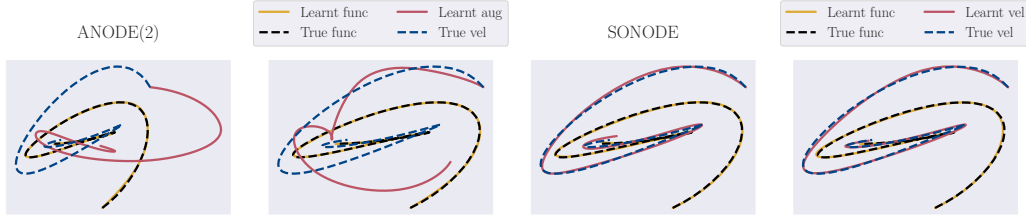

Figure 6: ANODEs and SONODEs successfully learn the trajectory in real space of a 2D ODE for two different random initialisations. However, the augmented trajectories of ANODE are in both cases widely different from the true velocity of the ODE *(left)*. In contrast, SONODE converges in both cases to the true ODE *(right)*.

the augmented trajectories differ greatly from the true velocity of the underlying ODE. In contrast, SONODE learns the correct velocity for both runs. This simple experiment confirms that ANODEs might not be a suitable investigative tool for scientific applications, where the physical interpretability of the results is important.

## 5.4 The functional loss landscape

The functional forms the two models converge to in Figure 6 are not a coincidence. Proposition 5.1 also has deeper implications for the ANODE's *(functional) loss landscape* when learning second order dynamics. Please refer to Appendix D for the proofs of the following results.

**Proposition 5.2.** *There are an infinity of (non-trivial) functional forms ANODEs can learn that model the true second order dynamics in real space.*

This means that there is an infinite number of functions ANODEs can approximate and obtain a zero loss. This suggests that an infinite number of global minima, representing different functions, may exist in the loss landscape of ANODEs. In contrast, we show that the second order constraints imposed on SONODE enforce that any global minima in its loss landscape approximate the same function — the acceleration and, in some cases, the initial velocity.

**Proposition 5.3.** *There is a unique functional form SONODEs can learn that models the true second order dynamics in real space.*

This is confirmed by our experiment from the previous section, where ANODE always converges to another augmented trajectory for each random initialisation (only two shown in the Figure 6), while SONODE always converges to the correct underlying ODE.

## 6 Experiments on second order dynamics

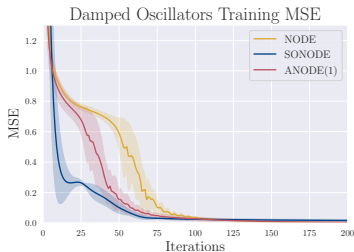

Figure 7: NODE, ANODE(1) and SONODE training on harmonic oscillators. SONODEs already have the second order behaviour built in as an architectural choice, so they are able to learn the dynamics in fewer iterations.

To test our above predictions, we perform an extensive comparison of ANODE and SONODE on a set of more challenging real and synthetic modelling tasks. These experiments provide further evidence for the described theoretical findings. Additional experimental details regarding the models and additional results are given in Appendix E.

### 6.1 Synthetic harmonic oscillators and noise robustness

**Harmonic oscillator** The most obvious application of SONODEs is on dynamical data from classical physics. This was tested by looking at a damped harmonic oscillator $\ddot{x} = -(\omega^2 + \gamma^2)x - 2\gamma\dot{x}$ with $\gamma = 0.1$ and $\omega = 1$ on 30 random pairs of initial positions and velocities. These were each evolved for 10 seconds, using one hundred evenly spaced time stamps. The loss depended on both position and velocity explicitly, therefore the models used the state $\mathbf{z} = [x, v]$ with the

option of augmentation for ANODEs. NODEs and ANODEs learnt a general $\dot{z}$, whereas SONODEs are given $\dot{z} = [v, f^{(a)}]$ and only learn $f^{(a)}$. SONODEs leverage their inductive bias and converge faster than the other models. Note that, all models were able to reduce the loss to approximately zero, as shown in Figure 7.

**Noise robustness**   We tested the models' abilities to learn a sine curve in varying noise regimes. The models were trained on fifty training points in the first ten seconds of $x = \sin(t)$, and then tested with ten points in the next five seconds. The train points had noise added to them, drawn from a normal distribution $\mathcal{N}(0, \sigma^2)$ for different standard deviations $\sigma = (0, 0.1, 0.2, \ldots, 0.7)$. The results presented in Figure 8 show that SONODEs are more robust to noise.

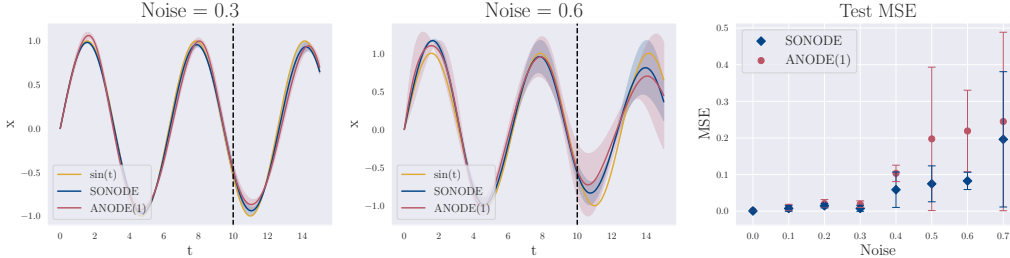

Figure 8: SONODEs and ANODEs learning a sine curve in different noise regimes: predictions for noise magnitude $0.3$ *(left)*, magnitude $0.6$ *(middle)* and test error for all noise levels *(right)*. The dotted line separates training and testing regimes. SONODEs are able to extrapolate better than ANODEs because they are forced to learn second order dynamics, and therefore are less likely to overfit the training points.

## 6.2   Experiments on real-world dynamical systems

**Airplane vibrations**   The dataset [16] concerns real vibrations measurements of an airplane. A shaker was attached underneath the right wing, producing an acceleration $a_1$. Additional accelerations at different points were measured including $a_2$, which was examined in this experiment, the acceleration on the right wing, next to a non-linear interface of interest. This is a higher order system, therefore it pertains to be a challenging modelling task. The results presented in Figure 9 show that while both methods can model the dynamics reasonably well, ANODEs perform marginally better. We conjecture that this is due to ANODEs not being restricted to second order behaviour, allowing them to partially access higher order dynamics. We test this conjecture in Appendix E.2.

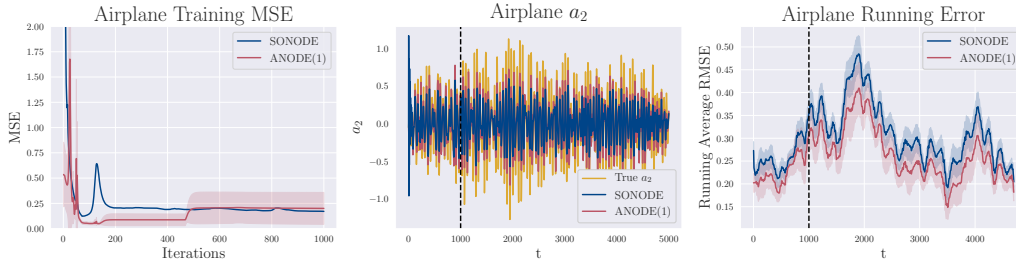

Figure 9: ANODE(1) and SONODE on the Airplane Vibrations dataset: training loss curves *(left)*, predicted value *(middle)*, and running error *(right)*. The models were trained on the first 1000 timestamps and then extrapolated to the next 4000. ANODEs are able to perform slightly better than SONODEs because they are able to access higher order dynamics.

**Silverbox oscillator**   The Silverbox dataset [21] is an electronic circuit resembling a Duffing Oscillator, with input voltage $V_1(t)$ and measured output $V_2(t)$. The non-linear model Silverbox represents is $\ddot{V}_2 = a\dot{V}_2 + bV_2 + cV_2^3 + dV_1$. To account for this, all models included a $V_2^3$ term. The results can be seen in Figure 10. On this second order system, SONODEs extrapolate better than ANODEs and are able to capture the increase in the amplitude of the signal exceptionally well.

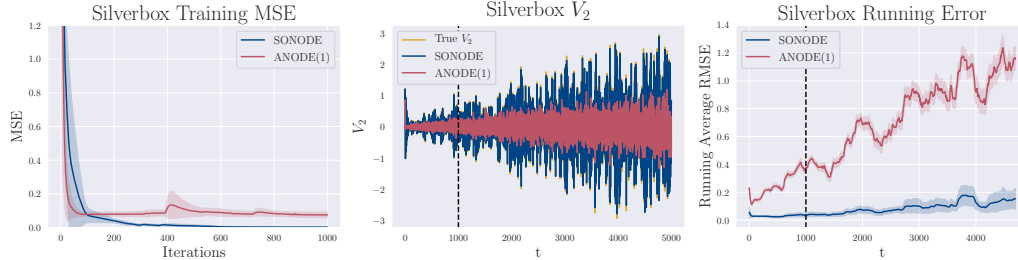

Figure 10: ANODE(1) and SONODE on the second order Silverbox dataset: training loss curves *(left)*, predicted value *(middle)*, and running error *(right)*. The models were trained on the first 1000 timestamps and extrapolated to the next 4000. As expected, SONODEs perform better.

# 7 Discussion and related work

**SONODEs vs ANODEs**   SONODEs can be seen as a special case of ANODEs, whose phase space dynamics are restricted to model second order behaviour. We believe that for tasks where the trajectory is unimportant, and performance depends only on the endpoints (such as classification), ANODEs might perform better because they are unconstrained in how they use their capacity (see Appendix E.4). In contrast, we expect SONODEs to outperform ANODEs both in terms of accuracy and convergence rate on time series data whose underlying dynamics is assumed (or known) to be second order. In this setting, SONODEs have a unique functional solution and fewer local minima compared to ANODEs. At the same time, they have higher parameter efficiency since $\dot{\mathbf{x}} = \mathbf{v}$ requires no parameters, so all parameters are in the acceleration. Finally, we expect SONODEs to be more appropriate for application in the natural sciences, where second order dynamics are common and it is useful to recover the force equation.

**Second Order Models**   Concurrent to our work, SONODEs have been briefly evaluated on MNIST by Massaroli et al. [14] as part of a wider study on Neural ODEs. In contrast, our study is focused on the theoretical understanding of second order behaviour. At the same time, our investigations are largely based on learning the dynamics of physical systems rather than classification tasks. Second order models have also been considered in Graph Differential Equations [17] and ODE$^2$VAE [24].

**Physics Based Models**   In the same way SONODEs assert Newtonian mechanics, other models have been made to use physical laws, guaranteeing physically plausible results, in discrete and continuous cases. Lutter et al. [13] apply Lagrangian mechanics to cyber-physical systems, while Greydanus et al. [6] and Zhong et al. [23] use Hamiltonian mechanics to learn dynamical data.

# 8 Conclusion

In this paper, we analysed how Neural ODEs (NODEs) can learn second order dynamics. In particular, we considered Second Order NODEs (SONODEs), a model constructed with this inductive bias in mind, and the more general class of Augmented Neural ODEs (ANODEs). We began by shedding light on the optimisation of SONODEs by generalising the adjoint sensitivity method from NODEs and comparing it with the training procedure of the equivalent coupled ODE. We also studied some of the theoretical properties of SONODEs and how they manifest in modelling toy problems.

We showed that, despite lacking the physics-based inductive biases of SONODEs, ANODEs are flexible enough to learn second order dynamics in practice. However, we also demonstrated, analytically and empirically, that they do this by learning to approximate an abstract coupled ODE where the state and augmented dimensions become entangled in the velocity. We proved that this has implications for interpretability in scientific applications as well as the 'shape' of the loss landscape. Our experiments on synthetic and real second order dynamical systems validate these concerns and reveal that the inductive biases of SONODE are generally beneficial in this setting.

Although this work investigates second order dynamics, the underlying principles of SONODEs can be readily extended to higher orders (a proof-of-principle is given in Appendix E.2). This, in turn, allows for modelling richer and more complex behaviour, while retaining the benefits of faster training and better modelling performance.

## Broader Impact

Neural ODEs are relatively new models and we are yet to see their full potential. We anticipate NODEs will see particular success in time-series data, which have a wide variety of real-world applications. Examples given by Jia and Benson [10] include the evolution of individuals' medical records and earthquake monitoring. Poli et al. [17] look at traffic forecasting and Greydanus et al. [6] show how a Neural ODE inspired by Hamiltonian mechanics can be applied to classical physics. Our work concerns Second Order Neural ODEs which can also be applied to classical physics, where Newton's second law describes the forces on an object.

Our theoretical work was concerned with demonstrating how best to use the adjoint method on SONODEs, and showing how the coupled ODE perspective of ANODEs leads to them being able to learn second order behaviour. Naturally, any impacts from this work will come from the applications of SONODEs.

We directly investigated two potential real-world applications of SONODEs. The Silverbox dataset, an electronic implementation of a damped spring with a non-linear spring constant. This naturally applies to circuits with oscillators, and damped elements, opening new directions to monitor circuits and signals. The dynamics can also be encountered in mechanical systems, including car suspension, which could be used to improve car safety. Note that, in our experiments, we also investigated the task of modelling the vibration dynamics of an aeroplane, which might lead to better and optimal aeroplane designs. Though contributions to civil mechanical engineering such as these have parallel applications in the design of weapons, it is not the case that our investigation expands technological capabilities in such a way as to enable new forms of warfare or to significantly improve current technologies (at this stage).

As stated, Neural ODEs are relatively new, and we are yet to see their full potential. We anticipate more applications to time series data in the future, which have many positive and negative applications, though at most we should think of our contribution as incremental in this regard and covered by existing institutions and norms.

## Acknowledgments and Disclosure of Funding

We would like to thank Cătălina Cangea, Jacob Deasy and Duo Wang for their helpful comments. We would like to also thank the reviewers for their constructive feedback and efforts towards improving our paper. The authors declare no competing interests.

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
