[Supplementary Material]

# A   Phase Space Trajectory Proofs

Here we present the proofs for the propositions from Section 4, concerning a $k$-th order initial value problem.

**Lemma A.1.** *For a k-th order IVP, where the k-th derivative is Lipschitz continuous, a solution cannot have discontinuities in the time derivative of its phase space trajectory.*

*Proof.* Consider the phase space trajectory $\mathbf{z}(t) = \left[ \mathbf{x}(t), \dfrac{d\mathbf{x}}{dt}(t), ..., \dfrac{d^{k-1}\mathbf{x}}{dt^{k-1}}(t) \right]$. Let $f$ be the k-th time derivative of $\mathbf{x}(t)$. Then the time derivative of $\mathbf{z}(t)$ is

$$
\frac{d}{dt}
\begin{bmatrix}
\mathbf{x} \\
\dfrac{d\mathbf{x}}{dt} \\
... \\
\dfrac{d^{k-1}\mathbf{x}}{dt^{k-1}}
\end{bmatrix}
=
\begin{bmatrix}
\dfrac{d\mathbf{x}}{dt} \\
\dfrac{d^2\mathbf{x}}{dt^2} \\
... \\
f(\mathbf{z})
\end{bmatrix}
$$

If for one set of finite arguments, $\mathbf{z}_1$, $f(\mathbf{z_1})$ is also finite, then because the gradients of $f$ are all bounded (due to Lipschitz continuity), for any other finite arguments, $\mathbf{z}_n$, $f(\mathbf{z}_n)$ will remain finite. Now consider $\dfrac{d^{k-1}\mathbf{x}}{dt^{k-1}}$, its time derivative is $f(\mathbf{z}(t))$, which is finite for all finite $\mathbf{z}$. Therefore, $\dfrac{d^{k-1}\mathbf{x}}{dt^{k-1}}$, can't have discontinuities with a finite derivative, and also must be finite for finite $\mathbf{z}$. Now consider $\dfrac{d^{k-2}\mathbf{x}}{dt^{k-2}}$, its time derivative is finite for all finite $\mathbf{z}$, and therefore it can't have discontinuities and also must be finite for all finite $\mathbf{z}$. This line of argument continues up to $\mathbf{x}$. The state $\mathbf{x}$ and all of its time derivatives up to the $k$-th have no discontinuities and are finite. Therefore as long as the initial conditions $\mathbf{z}(t_0)$ are finite, there can be no discontinuities in the time derivative of the phase space trajectory at finite time. $\square$

**Proposition 4.1.** *For a k-th order IVP, if the k-th derivative of $\mathbf{x}$ is Lipschitz continuous and has no explicit time dependence, then unique phase space trajectories cannot intersect at an angle. Similarly, a single phase space trajectory cannot intersect itself at an angle.*

*Proof.* Consider two trajectories $\mathbf{z}_1(t)$ and $\mathbf{z}_2(t)$ that have different initial conditions $\mathbf{z}_1(t_0) = \mathbf{h}_1$ and $\mathbf{z}_2(t_0) = \mathbf{h}_2$. Assume the trajectories cross at a point in phase space at an angle, $\mathbf{z}_1(t_1) = \mathbf{z}_2(t_2) = \tilde{\mathbf{h}}$. If they intersect at an angle, then evolving the two states by a small time $\delta t << 1$, and using the Lipschitz continuity of $f$, meaning that the trajectories cannot have kinks in them (as shown in Lemma A.1), $\mathbf{z}_1(t_1 + \delta t) \neq \mathbf{z}_2(t_2 + \delta t)$. However, if they are at the same point in phase space, then they must have the same k-th order derivative, $f$. All other derivatives are equal, so by evolving the states by the same small time $\delta t << 1$, $\mathbf{z}_1(t_1 + \delta t) = \mathbf{z}_2(t_2 + \delta t)$. There is a contradiction and therefore the assumption is wrong, unique trajectories cannot cross at an angle in phase space when $f$ is Lipschitz continuous and has no $t$ dependence.

Now consider the single trajectory $\mathbf{z}(t)$. Assume it intersects itself at an angle, at $t_1$ and $t_2$. Now consider two particles on this trajectory, starting at $t_1 - \tau$ and $t_2 - \tau$ such that $t_2 - \tau > t_1$. These two particles have different initial conditions and cross at an angle. However, the above shows that cannot happen. Therefore, the assumption that $\mathbf{z}(t)$ can intersect itself at an angle must be wrong. Trajectories cannot intersect themselves in phase space at an angle. $\square$

Trajectories can, however, feed into each other representing the same particle path at different times. Single phase space trajectories can feed into themselves representing periodic motion. This requires a Lipschitz continuous $f$, and for there to be no explicit time dependence. If there was time dependence then two trajectories can cross at different times, and a trajectory can self intersect. Effectively an additional dimension is added to phase space, which is time. The propositions above would still hold because $\dfrac{dt}{dt} = 1$ which is Lipschitz continuous. Therefore, with time included as a phase space dimension, intersections in space are only forbidden if they occur at the same time.

# B  Adjoint Sensitivity Method

We present a proof to both the first and second order Adjoint method, using a Lagrangian style approach [2, 5]. We also prove that when the underlying ODE is second order, using the first order method on a concatenated state, $\mathbf{z} = [\mathbf{x}, \mathbf{v}]$, produces the same results as the second order method but does so more efficiently. All parameters, $\theta$, are time-independent (so $\frac{d\theta}{dt} = \frac{dt}{d\theta} = 0$).

## B.1  First Order Adjoint Method

Let L denote a scalar loss function, $L = L(\mathbf{x}(t_n))$, the gradient with respect to a parameter $\theta$ is

$$\frac{dL}{d\theta} = \frac{\partial L}{\partial \mathbf{x}(t_n)^T} \frac{d\mathbf{x}(t_n)}{d\theta} \tag{14}$$

The vector $\dfrac{\partial L}{\partial \mathbf{x}(t_n)^T}$ is found using backpropagation. For dynamical data the loss will depend on multiple time stamps, there is also a sum over timestamps, $t_n$. Therefore $\dfrac{d\mathbf{x}(t_n)}{d\theta}$ is needed. $\mathbf{x}(t_n)$ follows

$$\mathbf{x}(t_n) = \int_{t_0}^{t_n} \dot{\mathbf{x}}(t) dt + \mathbf{x}(t_0) \tag{15}$$

subject to

$$\dot{\mathbf{x}} = f^{(v)}(\mathbf{x}, t, \theta_f), \qquad \mathbf{x}(t_0) = s(\mathbf{X}_0, \theta_s) \tag{16}$$

where $\mathbf{X}_0$ is the data going into the network and is constant. The functions $f^{(v)}$ and $s$ describe the ODE and the initial conditions. Here we allow $\mathbf{X}_0$ to first go through the transformation, $s(\mathbf{X}_0, \theta_s)$. This maintains generality and allows NODEs to be used as a component of a larger model. For example, $\mathbf{X}_0$ could go through a ResNet before the NODE, and then through a softmax classifier at the end (which is accounted for in the term $\dfrac{\partial L}{\partial \mathbf{x}(t_n)^T}$). Introduce the new variable $\mathbf{F}$

$$\mathbf{F} = \int_{t_0}^{t_n} \dot{\mathbf{x}}(t) dt = \int_{t_0}^{t_n} \left( \dot{\mathbf{x}} + A(t)(\dot{\mathbf{x}} - f^{(v)}) \right) dt + B(\mathbf{x}(t_0) - s) \tag{17}$$

These are equivalent because $(\dot{\mathbf{x}} - f^{(v)})$ and $(\mathbf{x}(t_0) - s)$ are both zero. This means the matrices, $A(t)$ and $B$, can be chosen freely (as long as they are well behaved, finite etc.), to make the computation easier. The gradients of $\mathbf{x}(t_n)$ with respect to the parameters are

$$\frac{d\mathbf{x}(t_n)}{d\theta_f} = \frac{d\mathbf{F}}{d\theta_f}, \qquad \frac{d\mathbf{x}(t_n)}{d\theta_s} = \frac{d\mathbf{F}}{d\theta_s} + \frac{ds(\mathbf{X}_0, \theta_s)}{d\theta_s} \tag{18}$$

Differentiating $\mathbf{F}$ with respect to a general parameter $\theta$

$$\frac{d\mathbf{F}}{d\theta} = \int_{t_0}^{t_n} \frac{d\dot{\mathbf{x}}}{d\theta} dt + \int_{t_0}^{t_n} A(t) \left( \frac{d\dot{\mathbf{x}}}{d\theta} - \frac{\partial f^{(v)}}{\partial \theta} - \frac{\partial f^{(v)}}{\partial \mathbf{x}^T} \frac{d\mathbf{x}}{d\theta} \right) dt + B \left( \frac{d\mathbf{x}(t_0)}{d\theta} - \frac{ds}{d\theta} \right) \tag{19}$$

Integrating by parts

$$\int_{t_0}^{t_n} A(t) \frac{d\dot{\mathbf{x}}}{d\theta} dt = \left[ A(t) \frac{d\mathbf{x}}{d\theta} \right]_{t_0}^{t_n} - \int_{t_0}^{t_n} \dot{A}(t) \frac{d\mathbf{x}}{d\theta} dt \tag{20}$$

Substituting this in and using $\int_{t_0}^{t_n} \frac{d\dot{\mathbf{x}}}{d\theta} dt = [\frac{d\mathbf{x}}{d\theta}]_{t_0}^{t_n}$, gives

$$\begin{aligned}
\frac{d\mathbf{F}}{d\theta} = &\left( \frac{d\mathbf{x}}{d\theta} + A(t) \frac{d\mathbf{x}}{d\theta} \right)\bigg|_{t_n} - \left( \frac{d\mathbf{x}}{d\theta} + A(t) \frac{d\mathbf{x}}{d\theta} \right)\bigg|_{t_0} - \int_{t_0}^{t_n} A(t) \frac{\partial f^{(v)}}{\partial \theta} dt \\
&- \int_{t_0}^{t_n} \left( \dot{A}(t) + A(t) \frac{\partial f^{(v)}}{\partial \mathbf{x}^T} \right) \frac{d\mathbf{x}}{d\theta} dt + B \left( \frac{d\mathbf{x}}{d\theta}\bigg|_{t_0} - \frac{ds}{d\theta} \right)
\end{aligned} \tag{21}$$

Using the freedom of choice of $A(t)$, let it follow the ODE

$$\dot{A}(t) = -A(t)\frac{\partial f^{(v)}}{\partial \mathbf{x}^T}, \qquad A(t_n) = -I \tag{22}$$

Where $I$ is the identity matrix. Then the first term and second integral in Equation (21) become zero, yielding

$$\frac{d\mathbf{F}}{d\theta} = (B - I - A(t_0)) \left.\frac{d\mathbf{x}}{d\theta}\right|_{t_0} + \int_{t_n}^{t_0} A(t)\frac{\partial f^{(v)}}{\partial \theta}dt - B\frac{ds}{d\theta} \tag{23}$$

Now using the freedom of choice of $B$, let it obey the equation

$$B = I + A(t_0) \tag{24}$$

This makes the first term in Equation (23) zero. This gives the final form of $\dfrac{d\mathbf{F}}{d\theta}$

$$\frac{d\mathbf{F}}{d\theta} = \int_{t_n}^{t_0} A(t)\frac{\partial f^{(v)}}{\partial \theta}dt - (I + A(t_0))\frac{ds}{d\theta} \tag{25}$$

Subbing into Equation (18) and using the fact that $f^{(v)}$ has no $\theta_s$ dependence and $s$ has no $\theta_f$ dependence

$$\frac{d\mathbf{x}(t_n)}{d\theta_f} = \int_{t_n}^{t_0} A(t)\frac{\partial f^{(v)}(\mathbf{x}, t, \theta_f)}{\partial \theta_f}dt, \qquad \frac{d\mathbf{x}(t_n)}{d\theta_s} = -A(t_0)\frac{ds(\mathbf{X}_0, \theta_s)}{d\theta_s} \tag{26}$$

This leads to the gradients of the loss

$$\frac{dL}{d\theta_f} = \frac{\partial L}{\partial \mathbf{x}(t_n)^T}\int_{t_n}^{t_0} A(t)\frac{\partial f^{(v)}(\mathbf{x}, t, \theta_f)}{\partial \theta_f}dt, \qquad \frac{dL}{d\theta_s} = -\frac{\partial L}{\partial \mathbf{x}(t_n)^T}A(t_0)\frac{ds(\mathbf{X}_0, \theta_s)}{d\theta_s} \tag{27}$$

Subject to the ODE for $A(t)$

$$\dot{A}(t) = -A(t)\frac{\partial f^{(v)}(\mathbf{x}, t, \theta_f)}{\partial \mathbf{x}}, \qquad A(t_n) = -I \tag{28}$$

Now introduce the adjoint state $\mathbf{r}(t)$

$$\mathbf{r}(t) = -A(t)^T\frac{\partial L}{\partial \mathbf{x}(t_n)}, \qquad \mathbf{r}(t)^T = -\frac{\partial L}{\partial \mathbf{x}(t_n)^T}A(t) \tag{29}$$

Using the fact that $\dfrac{\partial L}{\partial \mathbf{x}(t_n)}$ is constant with respect to time, the adjoint equations are obtained by applying the definition of the adjoint in Equation (29), to the gradients in Equation (27), and multiplying the ODE in Equation (28) by the constant $-\frac{\partial L}{\partial \mathbf{x}(t_n)}$

$$\frac{dL}{d\theta_f} = -\int_{t_n}^{t_0} \mathbf{r}(t)^T\frac{\partial f^{(v)}(\mathbf{x}, t, \theta_f)}{\partial \theta_f}dt, \qquad \frac{dL}{d\theta_s} = \mathbf{r}(t_0)^T\frac{ds(\mathbf{X}_0, \theta_s)}{d\theta_s} \tag{30}$$

Where the adjoint $\mathbf{a}(t)$ follows the ODE

$$\dot{\mathbf{r}}(t) = -\mathbf{r}(t)^T\frac{\partial f^{(v)}(\mathbf{x}, t, \theta_f)}{\partial \mathbf{x}}, \qquad \mathbf{r}(t_n) = \frac{\partial L}{\partial \mathbf{x}(t_n)} \tag{31}$$

The gradients are found by integrating the adjoint state, $\mathbf{r}$, and the real state, $\mathbf{x}$, backwards in time, which requires no intermediate values to be stored, using constant memory, a major benefit over traditional backpropagation.

These are the same equations that were derived by Chen et al. [3], however this includes the addition of letting $\mathbf{x}(t_0) = s(\mathbf{X}_0, \theta_s)$ giving the corresponding gradient, $\dfrac{dL}{d\theta_s}$. Additionally, the derivation used by Chen et al. [3] is simpler but does not present an obvious way to extend the adjoint method to second order ODEs, which this derivation method can do, as shown next.

## B.2 Second Order Adjoint

Using the same derivation method, but with a second order differential equation, a second order adjoint method is derived, according to the proposition from the main text:

**Proposition 3.1.** *The adjoint state* $\mathbf{r}(t)$ *of SONODEs follows the second order ODE*

$$\ddot{\mathbf{r}} = \mathbf{r}^T \frac{\partial f^{(a)}}{\partial \mathbf{x}} - \dot{\mathbf{r}}^T \frac{\partial f^{(a)}}{\partial \dot{\mathbf{x}}} - \mathbf{r}^T \frac{d}{dt}\left(\frac{\partial f^{(a)}}{\partial \dot{\mathbf{x}}}\right) \tag{32}$$

*and the gradients of the loss with respect to the parameters of the acceleration,* $\theta_f$ *are*

$$\frac{dL}{d\theta_f} = -\int_{t_n}^{t_0} \mathbf{r}^T \frac{\partial f^{(a)}}{\partial \theta_f} dt, \tag{33}$$

*Proof.* In general, the loss function, $L$, depends on $\mathbf{x}$ and $\dot{\mathbf{x}}$

$$\frac{dL}{d\theta} = \frac{\partial L}{\partial \mathbf{x}(t_n)^T}\frac{d\mathbf{x}(t_n)}{d\theta} + \frac{\partial L}{\partial \dot{\mathbf{x}}(t_n)^T}\frac{d\dot{\mathbf{x}}(t_n)}{d\theta} \tag{34}$$

The gradients from the positional part and the velocity part are found separately and added. Firstly the position

$$\mathbf{x}(t_n) = \int_{t_0}^{t_n} \dot{\mathbf{x}}(t)dt + \mathbf{x}(t_0) \tag{35}$$

Subject to the second order ODE

$$\ddot{\mathbf{x}} = f^{(a)}(\mathbf{x}, \dot{\mathbf{x}}, t, \theta_f), \qquad \mathbf{x}(t_0) = s(\mathbf{X}_0, \theta_s), \qquad \dot{\mathbf{x}}(t_0) = g(\mathbf{x}(t_0), \theta_g) \tag{36}$$

Following the same procedure as in first order, but including the initial condition for the velocity as well

$$\mathbf{F} = \int_{t_0}^{t_n} \dot{\mathbf{x}} + A(t)(\ddot{\mathbf{x}} - f^{(a)})dt + B(\dot{\mathbf{x}}(t_0) - g) + C(\mathbf{x}(t_0) - s) \tag{37}$$

As before, the vectors, $(\ddot{\mathbf{x}} - f^{(a)})$, $(\dot{\mathbf{x}}(t_0) - g)$ and $(\mathbf{x}(t_0) - s)$ are zero, which gives freedom to choose the matrices $A(t)$, $B$ and $C$ to make the calculation easier. The gradients of $\mathbf{x}(t_n)$ with respect to the parameters $\theta$ are

$$\frac{d\mathbf{x}(t_n)}{d\theta_f} = \frac{d\mathbf{F}}{d\theta_f}, \qquad \frac{d\mathbf{x}(t_n)}{d\theta_g} = \frac{d\mathbf{F}}{d\theta_g}, \qquad \frac{d\mathbf{x}(t_n)}{d\theta_s} = \frac{d\mathbf{F}}{d\theta_s} + \frac{ds(\mathbf{X}_0, \theta_s)}{d\theta_s} \tag{38}$$

Differentiating $F$ from equation 37 with respect to a general parameter

$$\begin{aligned}
\frac{d\mathbf{F}}{d\theta} &= \left[\frac{d\mathbf{x}}{d\theta}\right]_{t_0}^{t_n} - \int_{t_0}^{t_n} A(t)\frac{\partial f^{(a)}}{\partial \theta}dt + \int_{t_0}^{t_n} A(t)\left(\frac{d\ddot{\mathbf{x}}}{d\theta} - \frac{\partial f^{(a)}}{\partial \mathbf{x}^T}\frac{d\mathbf{x}}{d\theta} - \frac{\partial f^{(a)}}{\partial \dot{\mathbf{x}}^T}\frac{d\dot{\mathbf{x}}}{d\theta}\right)dt \\
&\quad + B\left(\left.\frac{d\dot{\mathbf{x}}}{d\theta}\right|_{t_0} - \frac{\partial g}{\partial \theta} - \frac{\partial g}{\partial \mathbf{x}(t_0)^T}\frac{d\mathbf{x}(t_0)}{d\theta}\right) + C\left(\left.\frac{d\mathbf{x}}{d\theta}\right|_{t_0} - \frac{ds}{d\theta}\right)
\end{aligned} \tag{39}$$

Integrating by parts

$$\int_{t_0}^{t_n} A(t)\frac{d\ddot{\mathbf{x}}}{d\theta}dt = \left[A(t)\frac{d\dot{\mathbf{x}}}{d\theta} - \dot{A}(t)\frac{d\mathbf{x}}{d\theta}\right]_{t_0}^{t_n} + \int_{t_0}^{t_n} \ddot{A}(t)\frac{d\mathbf{x}}{d\theta}dt \tag{40}$$

$$\int_{t_0}^{t_n} A(t)\frac{\partial f^{(a)}}{\partial \dot{\mathbf{x}}^T}\frac{d\dot{\mathbf{x}}}{d\theta}dt = \left[A(t)\frac{\partial f^{(a)}}{\partial \dot{\mathbf{x}}^T}\frac{d\mathbf{x}}{d\theta}\right]_{t_0}^{t_n} - \int_{t_0}^{t_n} \frac{d}{dt}\left(A(t)\frac{\partial f^{(a)}}{\partial \dot{\mathbf{x}}^T}\right)\frac{d\mathbf{x}}{d\theta}dt \tag{41}$$

Subbing these into Equation (39)

$$\frac{d\mathbf{F}}{d\theta} = \left[ \left( I - \dot{A} - A \frac{\partial f^{(a)}}{\partial \dot{\mathbf{x}}^T} \right) \frac{d\mathbf{x}}{d\theta} + A \frac{d\dot{\mathbf{x}}}{d\theta} \right]_{t_n} - \left[ \left( I - \dot{A} - A \frac{\partial f^{(a)}}{\partial \dot{\mathbf{x}}} \right) \frac{d\mathbf{x}}{d\theta} + A \frac{d\dot{\mathbf{x}}}{d\theta} \right]_{t_0}$$
$$+ \int_{t_0}^{t_n} \left( \ddot{A}(t) - A(t) \frac{\partial f^{(a)}}{\partial \mathbf{x}^T} + \frac{d}{dt} \left( A(t) \frac{\partial f^{(a)}}{\partial \dot{\mathbf{x}}^T} \right) \right) \frac{d\mathbf{x}}{d\theta} dt + \int_{t_n}^{t_0} A(t) \frac{\partial f^{(a)}}{\partial \theta} dt \qquad (42)$$
$$+ B \left( \frac{d\dot{\mathbf{x}}}{d\theta} \Big|_{t_0} - \frac{\partial g}{\partial \theta} - \frac{\partial g}{\partial \mathbf{x}(t_0)^T} \frac{d\mathbf{x}(t_0)}{d\theta} \right) + C \left( \frac{d\mathbf{x}}{d\theta} \Big|_{t_0} - \frac{ds}{d\theta} \right)$$

Using the freedom to choose $A(t)$, let it follow the second order ODE

$$\ddot{A}(t) = A(t) \frac{\partial f^{(a)}}{\partial \mathbf{x}^T} - \frac{d}{dt} \left( A(t) \frac{\partial f^{(a)}}{\partial \dot{\mathbf{x}}^T} \right), \qquad A(t_n) = 0, \qquad \dot{A}(t_n) = I \qquad (43)$$

This makes the first term and first integral in Equation (42) zero, yielding

$$\frac{d\mathbf{F}}{d\theta} = \int_{t_n}^{t_0} A(t) \frac{\partial f^{(a)}}{\partial \theta} dt + \left( \left( \dot{A}(t) + A(t) \frac{\partial f^{(a)}}{\partial \dot{\mathbf{x}}^T} - I - B \frac{\partial g}{\partial \mathbf{x}(t_0)^T} + C \right) \frac{d\mathbf{x}}{d\theta} \right) \Big|_{t_0}$$
$$+ \left( (B - A) \frac{d\dot{\mathbf{x}}}{d\theta} \right) \Big|_{t_0} - B \frac{\partial g}{\partial \theta} - C \frac{ds}{d\theta} \qquad (44)$$

Now using the freedom of choice in $B$ and $C$

$$B = A(t_0), \qquad C = -\dot{A}(t_0) - A(t_0) \frac{\partial f^{(a)}}{\partial \dot{\mathbf{x}}} \Big|_{t_0} + I + A(t_0) \frac{\partial g}{\partial \mathbf{x}(t_0)^T} \qquad (45)$$

This makes the second and third terms in Equation (44) zero, yielding

$$\frac{d\mathbf{F}}{d\theta} = \int_{t_n}^{t_0} A(t) \frac{\partial f^{(a)}}{\partial \theta} dt - B \frac{\partial g}{\partial \theta} - C \frac{ds}{d\theta} \qquad (46)$$

These give the final gradients of $\mathbf{x}(t_n)$ with respect to the parameters, by subbing the results for $B$, $C$ and $\frac{d\mathbf{F}}{d\theta}$ above into Equation (38), using the fact that $f^{(a)}$, $g$ and $s$ only depend on the parameters $\theta_f$, $\theta_g$ and $\theta_s$ respectively

$$\frac{d\mathbf{x}(t_n)}{d\theta_f} = \int_{t_n}^{t_0} A(t) \frac{\partial f^{(a)}}{\partial \theta_f} dt, \qquad \frac{d\mathbf{x}(t_n)}{d\theta_g} = -A(t_0) \frac{\partial g}{\partial \theta_g}$$
$$\frac{d\mathbf{x}(t_n)}{d\theta_s} = \left( \dot{A}(t_0) + A(t_0) \left( \frac{\partial f^{(a)}}{\partial \dot{\mathbf{x}}^T} \Big|_{t_0} - \frac{\partial g}{\partial \mathbf{x}(t_0)^T} \right) \right) \frac{ds}{d\theta_s} \qquad (47)$$

As before, introduce the adjoint state $\mathbf{r}^x(t)$:

$$\mathbf{r}^x(t) = -A(t)^T \frac{\partial L}{\partial \mathbf{x}(t_n)}, \qquad \mathbf{r}^x(t)^T = -\frac{\partial L}{\partial \mathbf{x}(t_n)^T} A(t) \qquad (48)$$

Using the fact that $\frac{\partial L}{\partial \mathbf{x}(t_n)}$ is constant with respect to time, all the results above, and the ODE and initial conditions for $A(t)$ in Equation (43) can be multiplied by $-\frac{\partial L}{\partial \mathbf{x}(t_n)^T}$, to get the gradients $\frac{dL}{d\theta}$ in terms of $\mathbf{r}^x(t)$

$$\frac{dL}{d\theta_f} = -\int_{t_n}^{t_0} \mathbf{r}^x(t)^T \frac{\partial f^{(a)}}{\partial \theta_f} dt, \qquad \frac{dL}{d\theta_g} = \mathbf{r}^x(t_0)^T \frac{\partial g}{\partial \theta_g}$$
$$\frac{dL}{d\theta_s} = \left( -\dot{\mathbf{r}}^x(t_0)^T - \mathbf{r}^x(t_0)^T \left( \frac{\partial f^{(a)}}{\partial \dot{\mathbf{x}}^T} \Big|_{t_0} - \frac{\partial g}{\partial \mathbf{x}(t_0)^T} \right) \right) \frac{d\mathbf{x}(t_0)}{d\theta_s} \qquad (49)$$

Subject to the second order ODE for $\mathbf{r}^x(t)$

$$\ddot{\mathbf{r}}^x(t) = \mathbf{r}^x(t)^T \frac{\partial f^{(a)}}{\partial \mathbf{x}} - \frac{d}{dt}\left(\mathbf{r}^x(t)^T \frac{\partial f^{(a)}}{\partial \dot{\mathbf{x}}}\right), \qquad \mathbf{r}^x(t_n) = 0, \qquad \dot{\mathbf{r}}^x(t_n) = -\frac{\partial L}{\partial \mathbf{x}(t_n)} \qquad (50)$$

Where after differentiating with the product rule the ODE in Equation (50) becomes

$$\ddot{\mathbf{r}}^x(t) = \mathbf{r}^x(t)^T \frac{\partial f^{(a)}}{\partial \mathbf{x}} - \dot{\mathbf{r}}^x(t)^T \frac{\partial f^{(a)}}{\partial \dot{\mathbf{x}}} - \mathbf{r}^x(t)^T \left(\frac{d}{dt}\frac{\partial f^{(a)}}{\partial \dot{\mathbf{x}}}\right) \qquad (51)$$

Where doing the full time derivative gives

$$\frac{d}{dt}\left(\frac{\partial f^{(a)}}{\partial \dot{\mathbf{x}}}\right) = [\dot{\mathbf{x}}^T, f^{(a)T}, 1]\begin{bmatrix}\frac{\partial}{\partial \mathbf{x}} \\ \frac{\partial}{\partial \dot{\mathbf{x}}} \\ \frac{\partial}{\partial t}\end{bmatrix}\left(\frac{\partial f^{(a)}}{\partial \dot{\mathbf{x}}}\right) \qquad (52)$$

Where the fact that $\ddot{x} = f^{(a)}$ has been used. This is only when the loss depends on the position. The same method is used to look at the velocity part in Equation (34)

$$\frac{dL}{d\theta} = \frac{\partial L}{\partial \dot{\mathbf{x}}(t_n)^T}\frac{d\dot{\mathbf{x}}(t_n)}{d\theta} \qquad (53)$$

Where

$$\dot{\mathbf{x}}(t_n) = \int_{t_0}^{t_n} \ddot{\mathbf{x}}(t)dt + \dot{\mathbf{x}}(t_0) \qquad (54)$$

The general method is to take this expression and add zeros, in the form of $A(t)$, $B$ and $C$ multiplied by the ODE and initial conditions, $(\ddot{\mathbf{x}} - f^{(a)})$, $(\dot{\mathbf{x}}(t_0) - g)$ and $(\mathbf{x}(t_0) - s)$. Then differentiate with respect to a general parameter $\theta$ and integrate by parts to get any integrals containing $\frac{d\dot{\mathbf{x}}}{d\theta}$ or $\frac{d\ddot{\mathbf{x}}}{d\theta}$ in terms of $\frac{d\mathbf{x}}{d\theta}$. Then choose the ODE for $A(t)$ to remove any $\frac{d\mathbf{x}}{d\theta}$ terms in the integral, and the initial conditions of $A(t_n)$ to remove the boundary terms at $t_n$. Then $B$ and $C$ are chosen to remove the boundary terms at $t_0$. After doing this the gradients of $\dot{\mathbf{x}}$ with respect to the parameters are

$$\frac{d\dot{\mathbf{x}}(t_n)}{d\theta_f} = \int_{t_n}^{t_0} A(t)\frac{\partial f^{(a)}}{\partial \theta_f}dt, \qquad \frac{d\dot{\mathbf{x}}(t_n)}{d\theta_g} = -A(t_0)\frac{\partial g}{\partial \theta_g}$$

$$\frac{d\dot{\mathbf{x}}(t_n)}{d\theta_s} = \left(\dot{A}(t_0) + A(t_0)\frac{\partial f^{(a)}}{\partial \dot{\mathbf{x}}^T}\bigg|_{t_0} - A(t_0)\frac{\partial g}{\partial \mathbf{x}(t_0)^T}\right)\frac{ds}{d\theta_s} \qquad (55)$$

Subject to the second order ODE for $A(t)$

$$\ddot{A}(t) = A(t)\frac{\partial f^{(a)}}{\partial \mathbf{x}^T} - \frac{d}{dt}\left(A(t)\frac{\partial f^{(a)}}{\partial \dot{\mathbf{x}}^T}\right), \qquad A(t_n) = -I, \qquad \dot{A}(t_n) = \frac{\partial f^{(a)}}{\partial \dot{\mathbf{x}}^T}\bigg|_{t_n} \qquad (56)$$

Now introduce the state $\mathbf{r}^v(t)$

$$\mathbf{r}^v(t) = -\frac{\partial L}{\partial \dot{\mathbf{x}}(t_n)^T}A(t), \qquad \mathbf{r}^v(t) = -A(t)^T\frac{\partial L}{\partial \dot{\mathbf{x}}(t_n)} \qquad (57)$$

Which allows the gradients of the loss with respect to the parameters to be written as

$$\frac{dL}{d\theta_f} = -\int_{t_n}^{t_0} \mathbf{r}^v(t)^T\frac{\partial f^{(a)}}{\partial \theta_f}dt, \qquad \frac{dL}{d\theta_g} = \mathbf{r}^v(t_0)^T\frac{\partial g}{\partial \theta_g}$$

$$\frac{dL}{d\theta_s} = \left(\mathbf{r}^v(t_0)^T\frac{\partial g}{\partial \mathbf{x}(t_0)^T} - \dot{\mathbf{r}}^v(t_0)^T - \mathbf{r}^v(t_0)^T\frac{\partial f^{(a)}}{\partial \dot{\mathbf{x}}^T}\bigg|_{t_0}\right)\frac{ds}{d\theta_s} \qquad (58)$$

Where $\mathbf{r}^v$ follows the second order ODE and initial conditions

$$\ddot{\mathbf{r}}^v(t) = \mathbf{r}^v(t)^T \frac{\partial f^{(a)}}{\partial \mathbf{x}} - \dot{\mathbf{r}}^v(t)^T \frac{\partial f^{(a)}}{\partial \dot{\mathbf{x}}} - \mathbf{r}^v(t)^T \frac{d}{dt}\left(\frac{\partial f^{(a)}}{\partial \dot{\mathbf{x}}}\right)$$

$$\mathbf{r}^v(t_n) = \frac{\partial L}{\partial \dot{\mathbf{x}}(t_n)}, \qquad \dot{\mathbf{r}}^v(t_n) = -\frac{\partial L}{\partial \dot{\mathbf{x}}(t_n)^T} \frac{\partial f^{(a)}}{\partial \dot{\mathbf{x}}}\bigg|_{t_n}$$

(59)

Now adding the gradients from the $\mathbf{x}$ dependence and the $\dot{\mathbf{x}}$ dependence together. It can be seen that the gradients are the same in Equations (49) and (58), but just swapping $\mathbf{r}^x$ and $\mathbf{r}^v$. Additionally, it can be seen from the ODEs for $\mathbf{r}^x$ and $\mathbf{r}^v$ in Equations (51) and (59), that they are governed by the same, linear, second order ODE, with different initial conditions. Therefore the gradients, $\frac{dL}{d\theta}$, can be written in terms of a new adjoint state, $\mathbf{r} = \mathbf{r}^x + \mathbf{r}^v$

$$\frac{dL}{d\theta_f} = -\int_{t_n}^{t_0} \mathbf{r}(t)^T \frac{\partial f^{(a)}(\mathbf{x}, \dot{\mathbf{x}}, t, \theta_f)}{\partial \theta_f} dt, \qquad \frac{dL}{d\theta_g} = \mathbf{r}(t_0)^T \frac{\partial g(\mathbf{x}(t_0), \theta_g)}{\partial \theta_g}$$

$$\frac{dL}{d\theta_s} = \left(\mathbf{r}(t_0)^T \frac{\partial g(\mathbf{x}(t_0), \theta_g)}{\partial \mathbf{x}(t_0)^T} - \dot{\mathbf{r}}(t_0)^T - \mathbf{r}(t_0)^T \frac{\partial f^{(a)}(\mathbf{x}, \dot{\mathbf{x}}, t, \theta_f)}{\partial \dot{\mathbf{x}}^T}\bigg|_{t_0}\right) \frac{ds(\mathbf{X}_0, \theta_s)}{d\theta_s}$$

(60)

Where $\mathbf{a}$ follows the second order ODE with initial conditions

$$\ddot{\mathbf{r}}(t) = \mathbf{r}(t)^T \frac{\partial f^{(a)}(\mathbf{x}, \dot{\mathbf{x}}, t, \theta_f)}{\partial \mathbf{x}} - \dot{\mathbf{r}}(t) \frac{\partial f^{(a)}(\mathbf{x}, \dot{\mathbf{x}}, t, \theta_f)}{\partial \dot{\mathbf{x}}} - \mathbf{r}(t)^T \frac{d}{dt}\left(\frac{\partial f^{(a)}(\mathbf{x}, \dot{\mathbf{x}}, t, \theta_f)}{\partial \dot{\mathbf{x}}}\right)$$

$$\mathbf{r}(t_n) = \frac{\partial L}{\partial \dot{\mathbf{x}}(t_n)}, \qquad \dot{\mathbf{r}}(t_n) = -\frac{\partial L}{\partial \mathbf{x}(t_n)} - \frac{\partial L}{\partial \dot{\mathbf{x}}(t_n)^T} \frac{\partial f^{(a)}(\mathbf{x}, \dot{\mathbf{x}}, t, \theta_f)}{\partial \dot{\mathbf{x}}}\bigg|_{t_n}$$

(61)

The full derivative, $d_t(\partial_{\dot{\mathbf{x}}} f^{(a)})$, is given by Equation (52). The ODE can also be written compactly as

$$\ddot{\mathbf{r}}(t) = \mathbf{r}(t)^T \frac{\partial f^{(a)}(\mathbf{x}, \dot{\mathbf{x}}, t, \theta_f)}{\partial \mathbf{x}} - \frac{d}{dt}\left(\mathbf{r}(t)^T \frac{\partial f^{(a)}(\mathbf{x}, \dot{\mathbf{x}}, t, \theta_f)}{\partial \dot{\mathbf{x}}}\right)$$

(62)

Just as in the first order method, a sum over times stamps $t_n$ may be required. This matches and extends on the gradients and ODE given by proposition 3.1. □

### B.3 Equivalence between the two Adjoint methods

When acting on a concatenated state, $\mathbf{z}(t) = [\mathbf{x}(t), \mathbf{v}(t)]$, the first order adjoint method will produce the same gradients as the second order adjoint method. However, it is more computationally efficient to use the first order method. This is also given in the main text as the following proposition:

**Proposition 3.2.** *The gradient of $\theta_f$ computed through the adjoint of the coupled ODE from (4) and the gradient from (6) are equivalent. However, the latter requires at least as many matrix multiplications as the former.*

Intuitively, the first order method will produce the same gradients because second order dynamics can be thought of as two coupled first order ODEs, where the first order dynamics happen in phase space. However, this provides no information about computational efficiency. We prove the equivalence and compare the computational efficiencies below.

*Proof.* The first order formulation of second order dynamics can be written as

$$\mathbf{z}(t) = \begin{bmatrix} \mathbf{x}(t) \\ \mathbf{v}(t) \end{bmatrix}, \qquad \dot{\mathbf{z}} = \begin{bmatrix} \mathbf{v} \\ f^{(a)}(\mathbf{x}, \mathbf{v}, t, \theta_f) \end{bmatrix}, \qquad \mathbf{z}(t_0) = \begin{bmatrix} \mathbf{x}(t_0) \\ \mathbf{v}(t_0) \end{bmatrix} = \begin{bmatrix} s(\mathbf{X}_0, \theta_s) \\ g(s(\mathbf{X}_0, \theta_s), \theta_g) \end{bmatrix}$$

(63)

When using index notation, $x_i$ and $v_i$ are concatenated to make $z_{i.}$. For $x_i$ and $v_i$, the index, i, ranges from 1 to $d$, whereas for $z_i$ it ranges from 1 to $2d$ accounting for the concatenation. This is

represented below

$$z_i = \begin{cases} x_i, & \text{if } i \leq d \\ v_{(i-d)}, & \text{if } i \geq d+1 \end{cases} \tag{64}$$

It also extends to $\dot{z}_i$ and $z_i(t_0)$, where $f_i^{(a)}$, $s_i$ and $g_i$ also have the index range from 1 to $d$, but the index of $\dot{z}_i$ goes from 1 to $2d$ just like for $z_i$.

$$\dot{z}_i = \tilde{f}_i^{(v)}(\mathbf{z}, t, \tilde{\theta}_f) = \begin{cases} v_i, & \text{if } i \leq d \\ f_{(i-d)}^{(a)}(\mathbf{x}, \mathbf{v}, t, \theta_f), & \text{if } i \geq d+1 \end{cases} \tag{65}$$

$$z_i(t_0) = \tilde{s}_i(\mathbf{X}_0, \tilde{\theta}_s) = \begin{cases} s_i(\mathbf{X}_0, \theta_s), & \text{if } i \leq d \\ g_{(i-d)}(s(\mathbf{X}_0, \theta_s), \theta_g), & \text{if } i \geq d+1 \end{cases} \tag{66}$$

Using the first order adjoint method, Equations (30) and (31), and using index notation with repeated indices summed over, the gradients are

$$\frac{dL}{d\tilde{\theta}_f} = -\int_{tn}^{t_0} r_i(t) \frac{\partial \tilde{f}_i^{(v)}(\mathbf{z}, t, \tilde{\theta}_f)}{\partial \tilde{\theta}_f} dt, \qquad \frac{dL}{d\tilde{\theta}_s} = r_i(t_0) \frac{d\tilde{s}_i(\mathbf{X}_0, \tilde{\theta}_s)}{d\tilde{\theta}_s} \tag{67}$$

Where the adjoint follows the ODE

$$\dot{r}_i(t) = -r_j(t) \frac{\partial \tilde{f}_j^{(v)}(\mathbf{z}, t, \tilde{\theta}_f)}{\partial z_i}, \qquad r_i(t_n) = \frac{\partial L}{\partial z_i(t_n)} \tag{68}$$

Where just like in $z_i$, the index, i, ranges from 1 to $2d$ in the adjoint $r_i(t)$. When writing the sum over the index explicitly

$$\dot{r}_i = -\sum_{j=1}^{2d} r_j \frac{\partial \tilde{f}_j^{(v)}}{\partial z_i} \qquad = -\sum_{j=1}^{d} r_j \frac{\partial \tilde{f}_j^{(v)}}{\partial z_i} - \sum_{j=d+1}^{2d} r_j \frac{\partial \tilde{f}_j^{(v)}}{\partial z_i} \tag{69}$$

Now split up the adjoint state, $\mathbf{r}$, into two equally sized vectors, $\mathbf{r}^A$ and $\mathbf{r}^B$, where their indices only range from 1 to $d$, like $\mathbf{x}$, $\mathbf{v}$, $f^{(a)}$, $g$ and $s$.

$$r_i = \begin{cases} r_i^A, & \text{if } i \leq d \\ r_{(i-d)}^B, & \text{if } i \geq d+1 \end{cases} \tag{70}$$

Using Equations (64), (65), (66) and (70), and subbing them into Equation (69), the derivative can be written as

$$\dot{r}_i = -\sum_{j=1}^{d} r_j^A \frac{\partial v_j}{\partial z_i} - \sum_{j=d+1}^{2d} r_{(j-d)}^B \frac{\partial f_{(j-d)}^{(a)}}{\partial z_i} \tag{71}$$

Relabelling the indices in the second sum $(j-d) \to j$

$$\dot{r}_i = -\sum_{j=1}^{d} r_j^A \frac{\partial v_j}{\partial z_i} - \sum_{j=1}^{d} r_j^B \frac{\partial f_j^{(a)}}{\partial z_i} \tag{72}$$

Looking at specific values of i:

$i \leq d$

$$\dot{r}_i = \dot{r}_i^A = -\sum_{j=1}^{d} r_j^A \frac{\partial v_j}{\partial x_i} - \sum_{j=1}^{d} r_j^B \frac{\partial f_j^{(a)}}{\partial x_i}, \qquad = -\sum_{j=1}^{d} r_j^B \frac{\partial f_j^{(a)}}{\partial x_i} \tag{73}$$

$i \geq d+1$

$$\dot{r}_i = \dot{r}_{(i-d)}^B = -\sum_{j=1}^{d} r_j^A \frac{\partial v_j}{\partial v_{(i-d)}} - \sum_{j=1}^{d} r_j^B \frac{\partial f_j^{(a)}}{\partial v_{(i-d)}} \tag{74}$$

Relabelling the first index $(i - d) \rightarrow i$

$$\dot{r}_i^B = -\sum_{j=1}^{d} r_j^A \frac{\partial v_j}{\partial v_i} - \sum_{j=1}^{d} r_j^B \frac{\partial f_j^{(a)}}{\partial v_i} \tag{75}$$

Noting that, $\frac{\partial v_j}{\partial v_i} = \delta_{ij}$, the time derivatives can be written in vector matrix notation as

$$\dot{\mathbf{r}}^A(t) = -\mathbf{r}^B(t)^T \frac{\partial f^{(a)}(\mathbf{x}, \mathbf{v}, t, \theta_f)}{\partial \mathbf{x}} \tag{76}$$

$$\dot{\mathbf{r}}^B(t) = -\mathbf{r}^A(t) - \mathbf{r}^B(t)^T \frac{\partial f^{(a)}(\mathbf{x}, \mathbf{v}, t, \theta_f)}{\partial \mathbf{v}} \tag{77}$$

Differentiating Equation (77), and using Equation (76) for $\dot{\mathbf{r}}^A(t)$

$$\ddot{\mathbf{r}}^B(t) = \mathbf{r}^B(t)^T \frac{\partial f^{(a)}(\mathbf{x}, \mathbf{v}, t, \theta_f)}{\partial \mathbf{x}} - \frac{d}{dt} \left( \mathbf{r}^B(t)^T \frac{\partial f^{(a)}(\mathbf{x}, \mathbf{v}, t, \theta_f)}{\partial \mathbf{v}} \right) \tag{78}$$

This matches the ODE for the second order method in Equation (62). Now applying the initial conditions, using index notation again

$$r_i(t_n) = \frac{\partial L}{\partial z_i(t_n)} \tag{79}$$

For $i \leq d$

$$r_i = r_i^A(t_n) = \frac{\partial L}{\partial x_i(t_n)} \tag{80}$$

For $i \geq d + 1$

$$r_i(t_n) = r_{(i-d)}^B(t_n) = \frac{\partial L}{\partial v_{(i-d)}(t_n)} \quad \rightarrow \quad r_i^B(t_n) = \frac{\partial L}{\partial v_i(t_n)} \tag{81}$$

Applying these initial conditions in $\mathbf{r}^A$ and $\mathbf{r}^B$ to Equation (77)

$$\dot{r}_i^B(t_n) = -\frac{\partial L}{\partial x_i(t_n)} - \frac{\partial L}{\partial v_j(t_n)} \frac{\partial f_j^{(a)}}{\partial v_i} \bigg|_{t_n} \tag{82}$$

By looking at the ODE and initial conditions, it is clear $\mathbf{r}^B$ is equivalent to the second order adjoint, in Equation (61). Now looking at the gradients, and including an explicit sum over the index

$$\frac{dL}{d\tilde{\theta}_f} = -\int_{t_n}^{t_0} \sum_{i=1}^{2d} r_i \frac{\partial \tilde{f}_i^{(v)}}{\partial \tilde{\theta}_f} dt \quad \rightarrow \quad = -\int_{t_n}^{t_0} \sum_{i=1}^{d} r_i^A \frac{\partial v_i}{\partial \tilde{\theta}_f} dt - \int_{t_n}^{t_0} \sum_{i=d+1}^{2d} r_{(i-d)}^B \frac{\partial f_{(i-d)}^{(a)}}{\partial \tilde{\theta}_f} dt \tag{83}$$

The first term is zero because $v$ has no explicit $\theta$ dependence. The second term, after relabelling and using summation convention becomes

$$\frac{dL}{d\theta_f} = -\int_{t_n}^{t_0} r_i^B(t) \frac{\partial f_i^{(a)}}{\partial \theta_f} dt \quad = -\int_{t_n}^{t_0} \mathbf{r}^B(t)^T \frac{\partial f^{(a)}}{\partial \theta_f} dt \tag{84}$$

Where $\tilde{\theta}_f = \theta_f$ has been used, as they are both the parameters for the acceleration. This matches the result for gradients of parameters in the acceleration term $\theta_f$, when using the second order adjoint method, because $\mathbf{r}^B$ is the adjoint.

Looking at the gradients related to the initial conditions

$$\frac{dL}{d\tilde{\theta}_s} = \mathbf{r}(t_0)^T \frac{d\tilde{s}(\mathbf{X}_0, \tilde{\theta}_s)}{d\tilde{\theta}_s} \tag{85}$$

After going through the previous process of separating out the sums from $1 \to d$ and $d+1 \to 2d$, then relabelling the indices on $\mathbf{r}^B$, this becomes

$$= r_i^A(t_0) \frac{ds_i(\mathbf{X}_0, \theta_s)}{d\tilde{\theta}_s} + r_i^B(t_0) \frac{dg_i(s(\mathbf{X}_0, \theta_s), \theta_g)}{d\tilde{\theta}_s} \tag{86}$$

Using the expression for $\mathbf{r}^A$ by rearranging Equation (77), this can be written as

$$\frac{dL}{d\tilde{\theta}_s} = \left( -\dot{r}_i^B(t_0) - r_j^B(t_0) \frac{\partial f_j^{(a)}}{\partial v_i} \Bigg|_{t_0} \right) \frac{ds_i}{d\tilde{\theta}_s} + r_i^B(t_0) \frac{dg_i}{d\tilde{\theta}_s} \tag{87}$$

The parameters $\tilde{\theta}_s$ contain both $\theta_s$ and $\theta_g$. Looking at $\theta_g$ first, where $s(\mathbf{X}_0, \theta_s)$ has no dependence

$$\frac{dL}{d\theta_g} = r_i^B(t_0) \frac{\partial g_i(s(\mathbf{X}_0, \theta_s), \theta_g)}{\partial \theta_g} = \mathbf{r}^B(t_0)^T \frac{\partial g(s(\mathbf{X}_0, \theta_s), \theta_g)}{\partial \theta_g} \tag{88}$$

where $\frac{dg}{d\theta_g}$ can be written as a partial derivative, because $\mathbf{X}_0$ and $\theta_s$ have no dependence on $\theta_g$. This expression is equivalent to $\frac{dL}{d\theta_g}$ found using the second order adjoint method. Now looking at the parameters $\theta_s$, these parameters are in $s(\mathbf{X}_0, \theta_s)$ explicitly and $g(s, \theta_g)$, implicitly through $s$. Subbing $\tilde{\theta}_s = \theta_s$ into Equation (87) gives

$$\frac{dL}{d\theta_s} = \left( -\dot{r}_i^B(t_0) - r_j^B(t_0) \frac{\partial f_j^{(a)}(\mathbf{x}, \mathbf{v}, t, \theta_f)}{\partial v_i} \Bigg|_{t_0} + r_j^B(t_0) \frac{\partial g_j(s(\mathbf{X}_0, \theta_s), \theta_g)}{\partial s_i} \right) \frac{ds_i(\mathbf{X}_0, \theta_s)}{d\theta_s} \tag{89}$$

Using the fact that $\mathbf{x}(t_0) = s$, this is the same result for $\frac{dL}{d\theta_s}$ found using the second order adjoint method:

$$\frac{dL}{d\theta_s} = \left( \mathbf{r}^B(t_0)^T \frac{\partial g(\mathbf{x}(t_0), \theta_g)}{\partial \mathbf{x}(t_0)^T} - \dot{\mathbf{r}}^B(t_0)^T - \mathbf{r}^B(t_0)^T \frac{\partial f^{(a)}(\mathbf{x}, \mathbf{v}, t, \theta_f)}{\partial \mathbf{v}^T} \Bigg|_{t_0} \right) \frac{ds(\mathbf{X}_0, \theta_s)}{d\theta_s} \tag{90}$$

All of the gradients match, so the first order adjoint method acting on $\mathbf{z}(t) = [\mathbf{x}(t), \mathbf{v}(t)]$ will produce the same gradients as the second order adjoint method acting on $\mathbf{x}(t)$. Given by Equation (60).

Looking at the efficiencies of each method and how they would be implemented. Both methods would integrate the state $\mathbf{z} = [\mathbf{x}, \mathbf{v}]$ forward in time, with $\dot{\mathbf{z}} = [\mathbf{v}, f^{(a)}]$. Both methods then integrate $\mathbf{z}$ and the adjoint backwards, in the same way. The difference is how the adjoint is represented. In first order it is represented as $[\mathbf{r}^A, \mathbf{r}^B]$ where $\mathbf{r}^B$ is the adjoint, in second order it is represented as $[\mathbf{r}, \dot{\mathbf{r}}]$ where $\mathbf{r}$ is the adjoint.

The time derivatives and initial conditions for the first order adjoint representation are

$$\begin{aligned}
\frac{d}{dt} \mathbf{r}^A(t) &= -\mathbf{r}^B(t)^T \frac{\partial f^{(a)}(\mathbf{x}, \mathbf{v}, t, \theta_f)}{\partial \mathbf{x}} \\
\frac{d}{dt} \mathbf{r}^B(t) &= -\mathbf{r}^A(t) - \mathbf{r}^B(t)^T \frac{\partial f^{(a)}(\mathbf{x}, \mathbf{v}, t, \theta_f)}{\partial \mathbf{v}} \\
\mathbf{r}^A(t_n) &= \frac{\partial L}{\partial \mathbf{x}(t_n)} \\
\mathbf{r}^B(t_n) &= \frac{\partial L}{\partial \mathbf{v}(t_n)}
\end{aligned} \tag{91}$$

The time derivatives and intial conditions for the second order adjoint representation are

$$\frac{d}{dt}\mathbf{r}(t) = \dot{\mathbf{r}}(t)$$

$$\frac{d}{dt}\dot{\mathbf{r}}(t) = \mathbf{r}(t)^T \frac{\partial f^{(a)}(\mathbf{x}, \mathbf{v}, t, \theta_f)}{\partial \mathbf{x}} - \dot{\mathbf{r}}(t)^T \frac{\partial f^{(a)}(\mathbf{x}, \mathbf{v}, t, \theta_f)}{\partial \mathbf{v}} - \mathbf{r}(t)^T \frac{d}{dt}\left(\frac{\partial f^{(a)}(\mathbf{x}, \mathbf{v}, t, \theta_f)}{\partial \mathbf{v}}\right)$$

$$\mathbf{r}(t_n) = \frac{\partial L}{\partial \mathbf{v}(t_n)}$$

$$\dot{\mathbf{r}}(t_n) = -\frac{\partial L}{\partial \mathbf{x}(t_n)} - \frac{\partial L}{\partial \mathbf{v}(t_n)^T} \frac{\partial f^{(a)}(\mathbf{x}, \mathbf{v}, t, \theta_f)}{\partial \mathbf{v}}\bigg|_{t_n}$$

(92)

Where

$$\frac{d}{dt}\left(\frac{\partial f^{(a)}}{\partial \mathbf{v}}\right) = [\mathbf{v}^T, f^{(a)T}, 1]\begin{bmatrix}\partial_{\mathbf{x}} \\ \partial_{\mathbf{v}} \\ \partial_t\end{bmatrix}\left(\frac{\partial f^{(a)}}{\partial \mathbf{v}}\right)$$

(93)

Looking at Equations (91) and (92), the second order method has the additional term, $\mathbf{r} \cdot d_t(\partial_{\mathbf{v}}(f^{(a)}))$, in the ODE, and the additional term, $(\partial_{\mathbf{v}}L) \cdot (\partial_{\mathbf{v}}f^{(a)})$ in the initial conditions. The first order method acting on the concatenated state, $[\mathbf{x}, \mathbf{v}]$, requires equal or fewer matrix multiplications than the second order method acting on $\mathbf{x}$, to find the gradients at each step and the initial conditions. This is in the general case, but also for all specific cases, it is as efficient or more efficient. The same is also true for calculating the final gradients. $\square$

The reason for the difference in efficiencies is the state, $\mathbf{r}^B$, is the adjoint, and the state, $\mathbf{r}^A$, contains a lot of the complex information about the adjoint. It is an entangled representation of the adjoint, contrasting with the disentangled second order representation $[\mathbf{r}, \dot{\mathbf{r}}]$. This is similar to how ANODEs can learn an entangled representation of second order ODEs and SONODEs learn the disentangled representation, seen in Section 5.3. However, entangled representations are more useful here, because they do not need to be interpretable, they just need to produce the gradients, and the entangled representation can do this more efficiently.

This analysis provides useful information on the inner workings of the adjoint method. It shows a second order specific method does exist, but the first order method acting on a state $\mathbf{z} = [\mathbf{x}, \mathbf{v}]$ will produce the same gradients more efficiently, due to how it represents the complexity. This was specific to second order ODEs, however, the first order adjoint will work on any system of ODEs, because any motion can be thought of as being first order motion in phase space. Additionally, the first order method may be the most efficient adjoint method. The complexity going from the first order to the second order was seen based on the calculation, so this is only likely to get worse as the system of ODEs becomes more complicated.

## C    Second Order ODEs are not Homeomorphisms

One of the conditions for a transformation to be a homeomorphism is for the transformation to be bijective (one-to-one and onto). In real space, a transformation that evolves according to a second order ODE does not have to be one-to-one. This is demonstrated using a one-dimensional counter-example

$$\ddot{x} = 0 \qquad \rightarrow \qquad x(t) = x_0 + v_0 t$$

$$x_0 = \begin{bmatrix}[0] \\ [1]\end{bmatrix}, \qquad v_0 = -x_0 + 2 = \begin{bmatrix}[2] \\ [1]\end{bmatrix}$$

If $t_0 = 0$ and $t_N = 1$

$$x(1) = \begin{bmatrix}[2] \\ [2]\end{bmatrix}$$

So the transformation in real space is not always one-to-one, and therefore, not always a homeomorphism.

# D ANODEs learning 2nd Order

Here we present the proofs for the propositions from Section 5

## D.1 Functional Form Proofs

**Proposition 5.1.** *The general form ANODEs learn second order behaviour is given by:*

$$\begin{bmatrix} \dot{\mathbf{x}} \\ \dot{\mathbf{a}} \end{bmatrix} = \begin{bmatrix} F(\mathbf{x}, \mathbf{a}, t, \theta_F) \\ G(\mathbf{x}, \mathbf{a}, t, \theta_G) \end{bmatrix}, \qquad G = \left( \frac{\partial F}{\partial \mathbf{a}^T} \right)^{-1}_{\text{left}} \left( f^{(a)} - \frac{\partial F}{\partial \mathbf{x}^T} F - \frac{\partial F}{\partial t} \right) \tag{94}$$

*Proof.* Let $\mathbf{z}(t)$ be the state vector $[\mathbf{x}(t), \mathbf{a}(t)]$. The time derivatives can be written as

$$\begin{bmatrix} \dot{\mathbf{x}}(t) \\ \dot{\mathbf{a}}(t) \end{bmatrix} = \begin{bmatrix} F(\mathbf{x}, \mathbf{a}, t, \theta_F) \\ G(\mathbf{x}, \mathbf{a}, t, \theta_G) \end{bmatrix} \tag{95}$$

Let $\mathbf{x}(t)$ follow the second order ODE, $\ddot{\mathbf{x}} = \dot{F} = f^{(a)}(\mathbf{x}, \dot{\mathbf{x}}, t, \theta_f)$. Differentiating $F$ with respect to time

$$\dot{F} = \frac{\partial F}{\partial \mathbf{x}^T} \dot{\mathbf{x}} + \frac{\partial F}{\partial \mathbf{a}^T} \dot{\mathbf{a}} + \frac{\partial F}{\partial t} = f^{(a)}(\mathbf{x}, \dot{\mathbf{x}}, t, \theta_f) \tag{96}$$

Using $\dot{\mathbf{x}} = F$ and $\dot{\mathbf{a}} = G$

$$f^{(a)}(\mathbf{x}, F, t, \theta_f) = \frac{\partial F}{\partial \mathbf{x}^T} F + \frac{\partial F}{\partial \mathbf{a}^T} G + \frac{\partial F}{\partial t} \tag{97}$$

Rearranging for G

$$G(\mathbf{x}, \mathbf{a}, t, \theta_G) = \left( \frac{\partial F}{\partial \mathbf{a}^T} \right)^{-1}_{\text{left}} \left( f^{(a)}(\mathbf{x}, F, t, \theta_f) - \frac{\partial F}{\partial \mathbf{x}^T} F - \frac{\partial F}{\partial t} \right) \tag{98}$$

$\square$

In order for the solution of $G$ to exist, the matrix $\dfrac{\partial F}{\partial \mathbf{a}^T}$ must be invertible. Either the dimension of $\mathbf{a}$ matches $F$, $\mathbf{x}$ and $f^{(a)}$, so that $\dfrac{\partial F}{\partial \mathbf{a}^T}$ is square, or $\dfrac{\partial F}{\partial \mathbf{a}^T}$ has a left inverse. Crucially, $F$ must have explicit $\mathbf{a}$ dependence, or the inverse does not exist. Intuitively, in order for real space to couple to augmented space, there must be explicit dependence.

Using the equation for $G(\mathbf{x}, \mathbf{a}, t, \theta_G)$, there is a gauge symmetry in the system, which proves proposition 5.2.

**Proposition 5.2.** *ANODEs can learn an infinity of (non-trivial) functional forms to learn the true dynamics of a second order ODE in real space.*

*Proof.* Assume a solution for $F(\mathbf{x}, \mathbf{a}, t, \theta_F)$ and $G(\mathbf{x}, \mathbf{a}, t, \theta_G)$ has been found such that, $\dot{F} = f^{(a)}$ and $F(\mathbf{x}_0, \mathbf{a}_0, t_0, \theta_F) = \dot{\mathbf{x}}_0$. If an arbitrary function of $\mathbf{x}$, $\phi(\mathbf{x})$, is added to $F$, where $\phi(\mathbf{x}_0) = 0$

$$\tilde{F}(\mathbf{x}, \mathbf{a}, t, \theta_F) = F(\mathbf{x}, \mathbf{a}, t, \theta_F) + \phi(\mathbf{x}) \tag{99}$$

The initial velocity is still the same. The dynamics are preserved if there is a corresponding change in $G$

$$\tilde{G}(\mathbf{x}, \mathbf{a}, t, \theta_G) = \left( \frac{\partial (F + \phi)}{\partial \mathbf{a}^T} \right)^{-1} \left( f^{(a)}(\mathbf{x}, F + \phi, t, \theta_f) - \frac{\partial (F + \phi)}{\partial \mathbf{x}^T} (F + \phi) - \frac{\partial (F + \phi)}{\partial t} \right) \tag{100}$$

The proof can end here, however this can be simplified. $\phi(\mathbf{x})$ has no explicit $\mathbf{a}$ or $t$ dependence, so this equation simplifies to

$$\tilde{G} = \left( \frac{\partial F}{\partial \mathbf{a}^T} \right)^{-1} \left( f^{(a)}(\mathbf{x}, F + \phi, t, \theta_f) - \frac{\partial F}{\partial \mathbf{x}^T} F - \frac{\partial F}{\partial t} - \frac{\partial F}{\partial \mathbf{x}^T} \phi - \frac{\partial \phi}{\partial \mathbf{x}^T} F - \frac{\partial \phi}{\partial \mathbf{x}^T} \phi \right) \tag{101}$$

The term $f^{(a)}(\mathbf{x}, F + \phi, t, \theta_f)$ can be Taylor expanded (assuming convergence)

$$f^{(a)}(\mathbf{x}, F + \phi, t, \theta_f) = f^{(a)}(\mathbf{x}, F, t, \theta_f) + \sum_{n=1}^{\infty} \left( \frac{\partial^n f^{(a)}(\mathbf{x}, \dot{\mathbf{x}}, t, \theta_f)}{\partial \dot{\mathbf{x}}^{Tn}} \bigg|_{\dot{\mathbf{x}}=F} \frac{\phi^n}{n!} \right) \quad (102)$$

Which gives the corresponding change in $G$

$$\tilde{G} = G(\mathbf{x}, \mathbf{a}, t, \theta_G) + \left( \frac{\partial F}{\partial \mathbf{a}^T} \right)^{-1} \left( \sum_{n=1}^{\infty} \left( \frac{\partial^n f^{(a)}}{\partial \dot{\mathbf{x}}^{Tn}} \bigg|_{\dot{\mathbf{x}}=F} \frac{\phi^n}{n!} \right) - \frac{\partial F}{\partial \mathbf{x}^T} \phi - \frac{\partial \phi}{\partial \mathbf{x}^T} F - \frac{\partial \phi}{\partial \mathbf{x}^T} \phi \right) \quad (103)$$

$\square$

This demonstrates that there are infinite functional forms that ANODEs can learn. This only considered perturbing functions $\phi(\mathbf{x})$. More complex functions can be added that have $\mathbf{a}$ or $t$ dependence, which lead to a more complex change in $G$. By contrast, we now show SONODEs have a unique functional form.

**Proposition 5.3.** *SONODEs learn to approximate a unique functional form to learn the true dynamics of a second order ODE in real space.*

*Proof.* Consider a dynamical system

$$\frac{d^2 \mathbf{x}}{dt^2} = f(\mathbf{x}, \mathbf{v}, t), \qquad \mathbf{x}(t_0) = \mathbf{x}_0, \qquad \mathbf{v}(t_0) = \mathbf{v}_0 \quad (104)$$

For these problems we let the loss only depend on the position, if it depends on position and velocity there would be more restrictions. So if it is true when loss only depends on the position, it is also true when it depends on both position and velocity.

Assume that there is another system, that has the same position as a function of time

$$\frac{d^2 \tilde{\mathbf{x}}}{dt^2} = \tilde{f}(\tilde{\mathbf{x}}, \tilde{\mathbf{v}}, t), \qquad \tilde{\mathbf{x}}(t_0) = \tilde{\mathbf{x}}_0, \qquad \tilde{\mathbf{x}}(t_0) = \tilde{\mathbf{v}}_0 \quad (105)$$

Where $f(\mathbf{x}, \mathbf{v}, t) \neq \tilde{f}(\tilde{\mathbf{x}}, \tilde{\mathbf{v}}, t)$. Because the initial conditions are given the position and velocity are defined at all times, and therefore position, velocity and acceleration can all be written as explicit functions of time. $\mathbf{x} \equiv \mathbf{x}(t)$, $\mathbf{v} \equiv \mathbf{v}(t)$. This allows for the acceleration to be written as a function of $t$ only, $f(\mathbf{x}, \mathbf{v}, t) = f_\tau(t)$ for all $t$. The same applies for the second system, $\tilde{\mathbf{x}} \equiv \tilde{\mathbf{x}}(t)$, $\tilde{\mathbf{v}} \equiv \tilde{\mathbf{v}}(t)$ and $\tilde{f}(\tilde{\mathbf{x}}, \tilde{\mathbf{v}}, t) = \tilde{f}_\tau(t)$

For all $t$, $\mathbf{x}(t) = \tilde{\mathbf{x}}(t)$, therefore, for any time increment, $\delta t$, $\mathbf{x}(t + \delta t) = \tilde{\mathbf{x}}(t + \delta t)$. Taking the full time derivative of $\mathbf{x}$ and $\tilde{\mathbf{x}}(t)$

$$\frac{d\mathbf{x}(t)}{dt} = \mathbf{v}(t) = \lim_{\delta t \to 0} \frac{\mathbf{x}(t + \delta t) - \mathbf{x}(t)}{\delta t} \quad (106)$$

$$\frac{d\tilde{\mathbf{x}}(t)}{dt} = \tilde{\mathbf{v}}(t) = \lim_{\delta t \to 0} \frac{\tilde{\mathbf{x}}(t + \delta t) - \tilde{\mathbf{x}}(t)}{\delta t} \quad (107)$$

Using these two equations and the fact that $\mathbf{x}(t) = \tilde{\mathbf{x}}(t)$, it is inferred that $\mathbf{v}(t) = \tilde{\mathbf{v}}(t)$ for all $t$. Taking the full time derivative of $\mathbf{v}(t)$ and $\tilde{\mathbf{v}}(t)$

$$\frac{d\mathbf{v}(t)}{dt} = f_\tau(t) = \lim_{\delta t \to 0} \frac{\mathbf{v}(t + \delta t) - \mathbf{v}(t)}{\delta t} \quad (108)$$

$$\frac{d\tilde{\mathbf{v}}(t)}{dt} = \tilde{f}_\tau(t) = \lim_{\delta t \to 0} \frac{\tilde{\mathbf{v}}(t + \delta t) - \tilde{\mathbf{v}}(t)}{\delta t} \quad (109)$$

Using these two equation and the fact that $\mathbf{v}(t) = \tilde{\mathbf{v}}(t)$ for all $t$, it is also inferred that $f_\tau(t) = \tilde{f}_\tau(t)$ for all $t$.

Using these three facts, $\mathbf{x}(t) = \tilde{\mathbf{x}}(t)$, $\mathbf{v}(t) = \tilde{\mathbf{v}}(t)$ and $f_\tau(t) = \tilde{f}_\tau(t)$. It must also be true that $f(\mathbf{x}(t), \mathbf{v}(t), t) = \tilde{f}(\tilde{\mathbf{x}}(t), \tilde{\mathbf{v}}(t), t) \rightarrow f(\mathbf{x}, \mathbf{v}, t) = \tilde{f}(\mathbf{x}, \mathbf{v}, t)$. Therefore the assumption that $f(\mathbf{x}, \mathbf{v}, t) \neq \tilde{f}(\tilde{\mathbf{x}}, \tilde{\mathbf{v}}, t)$ is incorrect, there can only be one functional form for $f(\mathbf{x}, \mathbf{v}, t)$.

Additionally, using $\mathbf{v}(t) = \tilde{\mathbf{v}}(t)$ for all $t$, the initial velocities must also be the same.

$\square$

### D.2 ANODEs Learning Two Functions

In Section 5.1, it was shown that ANODEs were able to learn two functions at the same time

$$x_1(t) = e^{-\gamma t}\sin(\omega t), \qquad x_2(t) = e^{-\gamma t}\cos(\omega t) \qquad (110)$$

using the solution

$$\begin{bmatrix} \dot{x} \\ \dot{a} \end{bmatrix} = \begin{bmatrix} Ca - \omega x - \gamma x + \omega \\ \omega a - \gamma a - \frac{1}{C}(2\omega^2 x + \gamma\omega - \omega^2) \end{bmatrix}, \qquad (111)$$

This is a specific case of the general formulation given by Equation (8). When the problem is generalised to have mixed amounts of sine and cosine in each function

$$x_1(t) = e^{-\gamma t}(A_1\sin(\omega t) + B_1\cos(\omega t)), \qquad x_2(t) = e^{-\gamma t}(A_2\sin(\omega t) + B_2\cos(\omega t)) \quad (112)$$

ANODEs are still able to learn these functions, shown in the first plot of Figure 11. As shown previously, if $F(\mathbf{x}, \mathbf{a}, t, \theta_F)$ gets the addition, $\alpha x + \beta$, then the ODE is preserved if $G(\mathbf{x}, \mathbf{a}, t, \theta_G)$ also gets the addition $\dfrac{-1}{C}((\alpha - \omega + \gamma)(\alpha x + \beta) + \alpha(Ca - \omega x - \gamma x + \omega))$, given by Equation (103). This gauge change preserves the ODE, but gives a new expression for the initial velocity

$$\dot{x}(0) = -\omega x(0) - \gamma x(0) + \omega + \alpha x(0) + \beta = \tilde{\alpha}x(0) + \tilde{\beta} \qquad (113)$$

which can be written in matrix-vector notation as

$$\begin{bmatrix} x_1(0) & 1 \\ x_2(0) & 1 \end{bmatrix} \begin{bmatrix} \tilde{\alpha} \\ \tilde{\beta} \end{bmatrix} = \begin{bmatrix} \dot{x}_1(0) \\ \dot{x}_2(0) \end{bmatrix} \qquad (114)$$

There are two equations and two unknowns, $\tilde{\alpha}$ and $\tilde{\beta}$, so this is possible to solve, and for ANODEs to learn.[2] To test this even further we added a third function to be learnt. ANODEs were able to do this, shown in the second plot of Figure 11.[3]

## E    Experimental Setup and Additional Results

We anticipate two main uses for SONODEs. One is using an experiment in a controlled environment, where the aim is to find values such as the coefficient of friction. The other use is when data is observed, and the aim is to extrapolate in time, but the experiment is not controlled, for example, observing weather. We would expect for the former, a simple model with only a single linear layer would be useful, to find those coefficients, and for the latter, a deeper model may be more appropriate. Additionally, Neural ODEs may be used in classification or other tasks that only involve the start and endpoints of the flow. For all of these tasks we used $t_0 = 0$ and $t_1 = 1$, and accelerations that were not time-dependent. For tasks depending on the start and endpoint only, a deeper neural network is more useful for the acceleration.

For all experiments, except the MNIST experiment, we optimise using Adam with a learning rate of 0.01. We also train on the complete datasets and do not minibatch. All the experiments were repeated 3 times to obtain a mean and standard deviation. Depending on the task at hand, we used two different architectures for NODEs, ANODEs and SONODEs. The first is a simple linear model,

Figure 11: ANODE(1) learning two functions *(left)* and three functions *(right)*, with a shared ODE, but different initial conditions. The real trajectories are seen going through their sampled data points, and the corresponding augmented trajectories are also plotted. ANODE(1) is able to learn the trajectories.

one weight matrix and bias without activations. This architecture, in the case of NODEs, ANODEs and SONODEs, was used on Silverbox, Airplane Vibrations and Van-Der-Pol Oscillator, with the aim of extracting coefficients from the models, for these tasks we also allowed ANODEs to learn the initial augmented position. The second architecture is a fully connected network with two hidden layers of size 20, it uses ELU activations in $\dot{z}$ and tanh activations in the initial conditions. ELU and tanh were used because they allow for negative values in the ODE [14].

When considering ANODEs, they are in a higher-dimensional space than the problem, and the result must be projected down to the lower dimensions. This projection was not learnt as a linear layer, instead, the components were directly selected, using an identity for the real dimensions, and zero for the augmented dimensions. This was done because a final (or initial) learnt linear layer would hide the advantages of certain models. For example, the parity problem can be solved easily if NODEs are given a final linear layer, do not move the points and then multiply by -1. For this reason, no models used a linear layer at the end of the flow. Equally, they do not initialise with a linear layer as they again hide advantages. For example, the nested n-spheres problem, NODEs can solve this with an initial linear layer, if they were to go into a higher-dimensional space the points may already be linearly separated, as shown by Massaroli et al. [14].

## E.1 Van Der Pol Oscillator

ANODEs and SONODEs were tested on a forced Van Der Pol (VDP) Oscillator that exhibits chaotic behaviour. More specifically, the parameters and equations of the particular VDP oscillator are:

$$\ddot{x} = 8.53(1 - x^2)\dot{x} - x + 1.2\cos(0.2\pi t), \qquad x_0 = 0.1, \qquad \dot{x}_0 = 0 \qquad (115)$$

As shown in Figure 12, while ANODEs achieve a lower training loss than SONODEs, their test loss is much greater. We conjecture that, in the case of ANODEs, this is a case of overfitting. SONODEs, on the other hand, can better approximate the dynamics, therefore they exhibit better predictive performance. Note that, neither model can learn the VDP oscillator particularly well, which may be attributed to chaotic behaviour of the system at hand.

## E.2 Third Order NODEs on Airplane Vibrations

We test Third Order Neural ODEs (TONODEs) on the Airplane Vibrations task from section 6.2. The results are in Figure 13.

We see that TONODEs vastly underperform compared to ANODEs and SONODEs. In each of the 3 repetitions of the experiment, the different initialisation found the best solution to be at zero. Therefore, whilst the loss stays constant, the error remains large. We hypothesise that despite

Figure 12: ANODE(1) and SONODE learning a Van-Der-Pol Oscillator: training loss curves *(left)*, predicted value *(middle)*, and running error *(right)*. The models were trained on the first 70 points and extrapolated to 200. ANODEs are able to converge to a lower training loss, however they diverge when extrapolating.

Figure 13: Repeating the Airplane Vibrations task with third order NODEs (TONODEs): training loss curves *(left)*, predicted value *(middle)*, and running error *(right)*. We see that, in this case, TONODEs are not as successful at modelling these dynamics as SONODEs and ANODEs, having a larger error both on the training data and the extrapolation.

theoretically being able to perform at least as well as SONODEs, TONODEs avoid exponentially growing at any point by exponentially decaying towards zero. It is likely that by rescaling the time to be between 0 and 1, TONODE would approach a more accurate solution.

### E.3 First Order Dynamics and Interpolation

SONODEs contain NODEs as a subset of models. Consider first order dynamics that is approximated by the NODE

$$\dot{\mathbf{x}} = f^{(v)}(\mathbf{x}, t, \tilde{\theta}_f) \tag{116}$$

Carrying out the full time derivative of Equation (116):

$$\ddot{\mathbf{x}} = \frac{\partial f^{(v)}(\mathbf{x}, t, \tilde{\theta}_f)}{\partial \mathbf{x}^T}\dot{\mathbf{x}} + \frac{\partial f^{(v)}(\mathbf{x}, t, \tilde{\theta}_f)}{\partial t}, \qquad \dot{\mathbf{x}}(t_0) = f^{(v)}(\mathbf{x}(t_0), t_0, \tilde{\theta}_f) \tag{117}$$

Which yields the SONODE equivalent of the learnt dynamics:

$$f^{(a)}(\mathbf{x}, \mathbf{v}, t, \theta_f) = \frac{\partial f^{(v)}(\mathbf{x}, t, \tilde{\theta}_f)}{\partial \mathbf{x}^T}\mathbf{v} + \frac{\partial f^{(v)}(\mathbf{x}, t, \tilde{\theta}_f)}{\partial t}, \qquad g(\mathbf{x}(t_0), \theta_g) = f^{(v)}(\mathbf{x}(t_0), t_0, \tilde{\theta}_f) \tag{118}$$

Additionally, it was shown in Equation (4) that SONODEs are a specific case of ANODEs that learn the initial augmented position. Therefore, anything that NODEs can learn, SONODEs should also be able to learn, and anything SONODEs can learn, ANODEs should be able to learn. To demonstrate that SONODEs and ANODEs can also learn first order dynamics, we task them with learning an exponential with no noise, $x(t) = exp(0.1667t)$. All models, as expected, are able to learn the function, as shown in Figure 14.

Figure 14: NODE *(left)*, ANODE(1) *(middle)* and SONODE *(right)* learning an exponential (simple first order dynamics) and interpolating between two observation sections. As expected, all models are able to learn the function.

## E.4 Performance on MNIST

NODEs, SONODEs and ANODEs were tested on MNIST [11] to investigate their ability on classification tasks. The networks used convolutional layers, which in the case of SONODEs were used for both the acceleration and the initial velocity. ANODEs were augmented with one additional channel as is suggested by Dupont et al. [4]. The models used a training batch size of 128 and test batch size of 1000, as well as group normalisation. SGD optimiser was used with a learning rate of 0.1 and momentum 0.9. The cross-entropy loss was used. The experiment was repeated 3 times with random initialisations to obtain a mean and standard deviation. The results are given in table 1 and Figure 15.

Table 1: Results for the MNIST experiments at convergence. SONODE converges to a higher test accuracy than NODEs with a lower NFE. ANODEs converge to the same higher test accuracy with a higher NFE, but with a lower parameter count than SONODEs.

| Model | Test Accuracy | NFE |
|---|---|---|
| NODE | $0.9961 \pm 0.0004$ | $26.2 \pm 0.0$ |
| SONODE | $\mathbf{0.9963 \pm 0.0001}$ | $\mathbf{20.1 \pm 0.0}$ |
| ANODE | $\mathbf{0.9963 \pm 0.0001}$ | $32.2 \pm 0.0$ |

In terms of test accuracy, SONODEs and ANODEs perform marginally better than NODEs. ANODEs can achieve the same accuracy with fewer parameters than SONODEs because the dynamics are not limited to second order and it is only the final state that is of concern in classification. However, SONODEs are able to achieve the same accuracy with a lower number of function evaluations (NFE). NFE denotes how many function evaluations are made by the ODE solver, and represents the complexity of the learnt solution. It is a continuous analogue of the depth of a discrete layered network. In the case of NODEs and ANODEs, the NFE gradually increases meaning that the complexity of the flow also increases. However, in the case of SONODEs, the NFE stays constant, suggesting that the initial velocity was associated with larger gradients (otherwise we would expect NFE to increase for SONODEs with training).

Figure 15: Comparing the performance of SONODEs and NODEs on the MNIST dataset: train accuracy *(left)*, test accuracy *(middle)*, NFE *(right)*. SONODEs converge to the same training accuracy and a higher test accuracy with a lower NFE than NODEs. NODEs had 208266 parameters, SONODEs had 283658 and ANODEs had 210626. Additional parameters were associated with the initial velocity, or the augmented channel.

## Footnotes

[2]There are trivial cases where this would be impossible. For example if the two functions were $\pm\sin(\omega t)$, they would have the same initial position, but different initial velocities. Corresponding to the matrix in Equation (114) having zero determinant.

[3]The figure also shows that when trajectories cross in real space they do not in augmented space, and when they cross in augmented space they do not in real space, supporting Proposition 4.1.