[Reviews · NeurIPS 2020]

Review 1

Summary and Contributions: The authors propose a neural ODE model for second order dynamics. The paper provides thorough theoretical and empirical analysis of both the representational capabilities and training dynamics of the model. In addition the authors also discuss how their proposed model is related to augmented neural ODEs and provide novel insights into the consequences of using augmentation for neural ODEs. Contributions: - The authors provide a nice theoretical analysis of the adjoint method for the case of second order ODEs. They show that directly using a second order adjoint method is equivalent to using a first order adjoint method on an augmented system but that the latter has lower computational cost. - The authors show that second order neural ODEs can overcome some of the representational limitations of neural ODEs in a similar way to augmented neural ODEs. However, the proposed method often finds “nicer” solutions than naive augmentation and provides a more interpretable understanding of what happens on the original (non augmented) space. - The authors also provide a thorough and interesting analysis of the effects of augmentation both theoretically and experimentally. The authors show that augmented models can learn higher order dynamics but that the learned representations are often “entangled” in the sense that the augmented dimensions do not exactly represent the velocity of the underlying second order system. - Through experiments on small scale physics datasets, the authors show that their proposed model performs better than regular augmented neural ODEs, particularly in the presence of noise and when the underlying dynamics are known to be exactly second order. Update after rebuttal: Thank you for the rebuttal. The rebuttal hasn't changed my score (mostly because I didn't think the paper had many issues to begin with). I think this is a strong paper and deserves to be accepted. I'm also excited to see the results of running 3rd order SONODEs on the airplane modeling task.

Strengths: Strengths: - The paper is extremely clear and thorough. The authors provide both theoretical and empirical justification for all their claims. - The motivation of the paper is interesting and provides a good first step towards neural ODE models that are appropriate for physical systems (or other second order systems). - The paper provides a nice new perspective on augmentation techniques in the case of neural ODEs. Significance: - I believe this paper provides important insights both for people interested in neural ODEs and for the machine learning + physics community. Novelty: - The methods and analysis provided in the paper are novel. Second order behaviour in neural ODEs has also been analysed in the dissecting neural ODEs paper, however it can be considered to be concurrent to this work. Further in this paper the focus is on physical systems whereas the dissecting neural ODEs paper focuses on classification.

Weaknesses: - The paper mostly focuses on quite small experiments. This is okay as the paper is mostly concerned with analysing the higher order behaviour of neural ODEs and getting a detailed understanding of this. However, it would be nice to see larger scale experiments if possible. Are there any larger scale physics (or other second order) datasets that could be tested? What about modeling e.g. n-particle dynamics? - The proposed model is limited to second order behaviour even though higher order behaviour is not uncommon in real life physical systems.

Correctness: The claims and methods in the paper are, to the best of my understanding, correct. The empirical methodology is also correct and thorough. I appreciate that almost all experiments contain error bars and standard deviations across several runs. The authors also provide the code to reproduce all experiments which is great.

Clarity: The paper is extremely well written. I like that the authors often start with a motivating example to give intuition and then move on to more general statements (e.g. section 5.1), which makes the paper clear and easy to follow. The figures are also great and the proofs in the appendix are clear and well explained.

Relation to Prior Work: The authors provide a good and clear discussion about how their work is related to previous contributions. A similar model for second order dynamics is given in the dissecting neural ODEs paper, however this can be considered concurrent work and further, as the authors mention, the dissecting neural ODEs paper focuses mostly on empirical classification tasks whereas this paper provides a deeper analysis and understanding of this behaviour, particularly in the context of physical systems and augmentation.

Reproducibility: Yes

Additional Feedback: As you mentioned in section 6.2, some physical systems have behaviour that is order 3 or higher. Have you considered trying to model higher order behaviour with your model? Wouldn’t it be fairly simple to extend this model into a 3 * d dimensional phase space and restrict the dynamics function appropriately? Would this potentially improve performance on the airplane vibrations task? The broader impact statement is particularly well thought through and I appreciate that the authors took the time to consider this in detail. Typo in line 695 in appendix: grater -> greater


Review 2

Summary and Contributions: This paper proposes second order neural ODEs, a constrained version of augmented neural ODEs, that can sometimes facilitate learning of time series better than the unconstrained augmented neural ODE version. They derive new methods for training second order neural ODEs in the process.

Strengths: The progression of ideas from neural ODEs to second-order neural ODEs is quite clear. The propositions and numerical experiments are as comprehensive as one would expect for a ten-page paper. The appendix is well-explained and comprehensive as well. I particularly appreciate that the authors showed an example of the less constrained augmented neural ODE approach beating their second order neural ODE approach (in Fig. 8) because it illustrates that the constraints that they use will not be appropriate for certain types of time series.

Weaknesses: Honestly, the only weakness of this paper is that the constraints being placed on ANODE to get SONODE can be limiting. Despite new experiments, this remains my worry.

Correctness: Yes, and yes.

Clarity: Yes.

Relation to Prior Work: Yes.

Reproducibility: Yes

Additional Feedback:


Review 3

Summary and Contributions: This paper focuses on the second order neural ODEs and shows that the adjoint sensitivity method can be extended to this framework, furthermore, an alternative first order optimization method is computationally more efficient.

Strengths: The extension from first order ODE to second order has application potential. Simulations are clear and indicate what is expected.

Weaknesses: I am not familiar with the background of this paper, NODEs. But at high level, it seems to be technically trivial to reduce a higher order ODE to a first order one. I hope the author could have provide a brief discussion in introduction on the technical difficulties in this paper. A more complete background in residual nets might be helpful as well. In line 28 of Introduction, "To fill this void,....", this is where I want to see the necessity of filling this void and the potential impact of the results of this paper to this area. It is pointed that "no general study of second order behaviour neural ODE...", but please address the meaning of this study for machine learning community and the technical novelty. The paper is expected to benefit the general audience, not only the experts in specific narrow area. But with the current write up, I am not convinced that it has given an important result or introduced an interesting technique that might be used in other problems.

Correctness: There is no obvious mistakes in proofs and calculations.

Clarity: The paper is clearly structured with preliminary, theory and experiments, but more details on the non-triviality of the ODE technique and the importance of the results are required to be clarified in introduction.

Relation to Prior Work: This is my main concern, the authors are expected to give a detailed comparison to the previous work by addressing: why the result matters and why the technique is difficult.

Reproducibility: Yes

Additional Feedback:


Review 4

Summary and Contributions: The paper proposes the second order neural ODE (SONODE) that can describe higher order dynamics and tackle the intersecting trajectory problem of NODE. It alsoinvestigates and its relationship to augmented neural ODE (ANODE) theoretically and empirically, and extends the adjoint method to SNONODE. It shows that SONODE has unique solution to the problem that ANODE has non-unique solutions. The method was tested on synthesized and real data with the comparison to ANODE.

Strengths: The paper provides theoretical understanding of the relationship between SONODE and ANODE. SONODE tackles the issue of intersecting trajectory to first order ODEs and gives unique solution to certain problems that ANODE cannot.

Weaknesses: The proposed SONODE can be seen as a special case of ANODE of which the augmented state is constrained to be the second order derivative (acceleration). Its functional form f^(a) takes functional form df^(v)/dt if I am correct. The can you talk about if SONODE can represent the function used in Dupont2019 as the example showing ANODE's advantage? Moreover ANODE has a larger parameter space than SONODE. Then woulde ANODE be able to achieve the same performance with adequate amount of data? The authors have addressed these concerns in the feedback. It is agreed that SONODE is a special case of ANODE.

Correctness: The claims, method and the empirical methodology are correct to the best of my knowledge.

Clarity: The paper is overall well written. It's a little confusing that the variable 'a' stands for both acceleration and augmented variable. Can you please explain more on how Eq 8. is derived? To me it is a jump from \dot{z} = [\dot{x}, \dot{a}] to what it is in Eq 8. Thank the authors for clarifying the derivation.

Relation to Prior Work: It is clearly discussed how this work differs from previous contributions

Reproducibility: Yes

Additional Feedback: The supplementary compares SONODE and NODE on classification problem. Can you please do the same comparison with ANODE.

[Author Response · NeurIPS 2020]

We thank the reviewers for their time and thoughtful feedback. Overall, the reviewers thought the paper was well
written (**R1**, **R2**, **R4**) and found the theoretical analysis interesting (**R1**, **R4**). They thought the empirical methodology
was comprehensive (**R2**) and thorough (**R1**), and they recognised that Second Order Neural ODEs (SONODEs) have
application potential (**R3**). We are very grateful for the suggestions on how to improve this work. A key point is that the
paper would benefit from further comparisons (theoretical and empirical) between Augmented Neural ODEs (ANODEs)
and SONODEs, and a discussion about the settings in which ANODEs are expected to outperform SONODEs and vice
versa (**R2**, **R4**). We agree that this would benefit the work, therefore we address this concern and others below. We
refer to lines, figures and pages from the manuscript as (L, Fig., p).

**Make further comparisons between ANODEs and SONODEs (R2, R4)** - Given SONODEs are a special case of
ANODEs (Eq.4, L65), the reviewers ask when ANODEs might outperform SONODEs (**R2**) and if ANODEs can
achieve the same performance with adequate data (**R4**). This depends on the task and the expected underlying dynamics.
We believe that for tasks where the trajectory is unimportant, and performance depends only on the end points (such
as classification), ANODEs might perform better because they are unconstrained in how they use their capacity. To
investigate this, we followed **R4**'s suggestion and included ANODEs in the MNIST experiment (as we did for NODEs
and SONODEs in Appendix E.3), augmenting along the channels as is done in Dupont 2019. We found that ANODEs
achieve the same accuracy as SONODEs (see figure below), with fewer parameters and only one augmented channel.
This is consistent with the result from Dissecting Neural ODEs, where ANODEs had a higher accuracy with five
augmented channels and approximately the same number of parameters.

19 
Figure 1: MNIST evaluation: test accuracy and number of function evaluations (NFE).

We expect SONODEs to outperform ANODEs on time-series data when the underlying dynamics is assumed (or known) to be second order (also mentioned by **R1**). In this setting, SONODEs have a unique functional solution and fewer local minima compared to ANODEs. For example, in theory ANODEs can learn the Silverbox task but they are unable to do so (Fig.9). Moreover, better interpretability also makes SONODEs more appropriate for application in the natural sciences, where second order dynamics are common and it is useful to recover the force equation. Additionally, SONODEs train faster as they do not have to learn second order (Fig.6), they are
31 more robust to noise (Fig.7), and will require fewer parameters ($\dot{x} = v$ does not require any parameters). *However*,
32 when the dynamics are not second order we believe ANODEs will perform better as they are not restricted to second
33 order solutions as shown in Fig.8 (the airplane benchmark).

34 **More complex systems, e.g. n-particle dynamics, and higher-order dynamics (R1)** - Our approach allows for
35 modeling more complex systems, such as n-particle dynamics. For instance, if $\mathbf{x} = (\mathbf{x}_1, ..., \mathbf{x}_n)$ and $\mathbf{v} = (\mathbf{v}_1, ..., \mathbf{v}_n)$;
36 then $\dot{\mathbf{x}} = \mathbf{v}, \dot{\mathbf{v}} = f^{(a)}(\mathbf{x}, \mathbf{v}, t, \theta_f), \mathbf{v}(t_0) = g(\mathbf{x}(t_0), t_0, \theta_g)$ models the dynamics. However, even simple multi-particle
37 systems are highly sensitive to initial conditions (chaotic) and it becomes computationally intractable to solve the
38 problem to acceptable precision. Moreover, while in this paper we investigate second order dynamics, SONODEs can
39 indeed be extended to higher order to model richer behaviour. We will investigate this by comparing third order to
40 SONODEs and ANODEs on more difficult modelling tasks, such as the airplane task.

41 **Motivation for SONODEs and detailed comparison to previous work (R3)** - The paper focuses on the theoretical
42 and empirical analysis of second order behaviour in ANODEs. We note that second order dynamics are common
43 in physics (L27-28), however, we will amend the introduction to better contextualise the importance of this work.
44 Moreover, while we discuss most of the relevant related work, we will consider extending this discussion to include a
45 broader overview of related work.

46 **Can SONODEs represent the function used in Dupont et al. 2019? (R4)** - Yes, we demonstrate SONODEs on the
47 $g_{1d}$ and g (p3,p4). However, we use the names compact parity problem (as we consider the generalised parity problem),
48 and nested-n-spheres (name used in Dissecting Neural ODEs). We will add a note on this in the final version.

49 **Confusing that 'a' stands for both acceleration and augmented variable (R4)** - Whilst we use only 'a' implicitly in
50 the function $f^{(a)}$, we agree with the meta-point that, in dynamics, 'a' often refers to acceleration, which is very relevant
51 in this work. We will amend the text to use a different symbol for the augmented variable to remove this confusion.

52 **Can you explain how Eq 8. is derived? (R4)** - Start from the state $\mathbf{z} = [\mathbf{x}, \mathbf{a}]$. The velocity can almost be represented
53 by $\mathbf{a}$, but in the original formulation $\mathbf{a}(t_0) = 0$. This is fixed by adding the *constant* $\dot{\mathbf{x}}(t_0)$. Such that $\dot{\mathbf{x}} = \mathbf{a} + \dot{\mathbf{x}}(t_0)$.
54 To get the desired acceleration, $\ddot{\mathbf{x}} = \dot{\mathbf{a}} = f^{(a)}(\mathbf{x}, \mathbf{a} + \dot{\mathbf{x}}(t_0), t, \theta_f)$. Using $\dot{\mathbf{x}}(t_0) = g(\mathbf{x}(t_0), t_0, \theta_g)$ from the SONODE
55 formulation gives Eq 8. This is consistent with the more general expression in Eq 12.

[Meta-Review · NeurIPS 2020]

Four expert reviewers provided insightful reviews. This is a very well written paper with a nice idea incrementally improving the NODE/ANODE framework. When the dynamical system is well approximated by 2nd order dynamics, it makes perfect sense to use it to limit the expressive power and gain in computation. I recommend acceptance. Please consider incorporating the rebuttal into the final paper.